# Presenting native-like trimeric HIV-1 antigens with self-assembling nanoparticles

Linling He[1,*], Natalia de Val[2,3,4,*], Charles D. Morris[1], Nemil Vora[1], Therese C. Thinnes[1], Leopold Kong[2,3,4], Parisa Azadnia[1], Devin Sok[1,3,4], Bin Zhou[5], Dennis R. Burton[1,3,4,6], Ian A. Wilson[2,3,4,7,8], David Nemazee[1], Andrew B. Ward[2,3,4] & Jiang Zhu[1,2,4]

Structures of BG505 SOSIP.664 trimer in complex with broadly neutralizing antibodies (bNAbs) have revealed the critical role of trimeric context for immune recognition of HIV-1. Presentation of trimeric HIV-1 antigens on nanoparticles may thus provide promising vaccine candidates. Here we report the rational design, structural analysis and antigenic evaluation of HIV-1 trimer-presenting nanoparticles. We first demonstrate that both V1V2 and gp120 can be presented in native-like trimeric conformations on nanoparticles. We then design nanoparticles presenting various forms of stabilized gp140 trimer based on ferritin and a large, 60-meric E2p that displays 20 spikes mimicking virus-like particles (VLPs). Particle assembly is confirmed by electron microscopy (EM), while antigenic profiles are generated using representative bNAbs and non-NAbs. Lastly, we demonstrate high-yield gp140 nanoparticle production and robust stimulation of B cells carrying cognate VRC01 receptors by gp120 and gp140 nanoparticles. Together, our study provides an arsenal of multivalent immunogens for HIV-1 vaccine development.

[1] Department of Immunology and Microbial Science, The Scripps Research Institute, 10550 N Torrey Pines Road, La Jolla, California 92037, USA. [2] Department of Integrative Structural and Computational Biology, The Scripps Research Institute, La Jolla, California 92037, USA. [3] International AIDS Vaccine Initiative Neutralizing Antibody Center and the Collaboration for AIDS Vaccine Discovery, The Scripps Research Institute, La Jolla, California 92037, USA. [4] Scripps Center for HIV/AIDS Vaccine Immunology and Immunogen Discovery, The Scripps Research Institute, La Jolla, California 92037, USA. [5] Department of Chemistry, The Scripps Research Institute, La Jolla, California 92037, USA. [6] Ragon Institute of Massachusetts General Hospital, Massachusetts Institute of Technology and Harvard, Cambridge, Massachusetts 02139-3583, USA. [7] The Joint Center for Structural Genomics, The Scripps Research Institute, La Jolla, California 92037, USA. [8] Skaggs Institute for Chemical Biology, The Scripps Research Institute, La Jolla, California 92037, USA. * These authors contributed equally to this work. Correspondence and requests for materials should be addressed to A.B.W. (email: abward@scripps.edu) or to J.Z. (email: jiang@scripps.edu).

A critical goal of vaccine development for human immunodeficiency virus type-1 (HIV-1) is to induce broadly neutralizing antibodies (bNAbs) in naïve individuals[1]. Diverse bNAb families have been identified from HIV-1-infected individuals[2–4], revealing multiple sites of HIV-1 vulnerability on the envelope (Env) glycoprotein. The functional Env is a trimer of heterodimers, each containing a receptor-binding protein (gp120) and a transmembrane fusion protein (gp41), which associate into a viral spike via non-covalent interactions[5]. This trimeric spike is inherently labile, which has hindered rational vaccine design due to a limited structural understanding of Env. The BG505 SOSIP.664 gp140 trimer[6] has provided an excellent antigenic[7,8] and structural[9–11] mimic of the native spike. Structures of this trimer bound to various bNAbs illustrated the critical role of trimeric context in the recognition of Env by humoral responses[9,10,12–17]. Following the development of cleaved SOSIP trimers[18–21], cleavage-independent, well-folded gp140 trimers were also proposed as alternative trimer immunogens[22,23].

Soluble trimer alone, however, may not be the optimal platform for HIV-1 vaccines, because subunit vaccines are often not as immunogenic as those based on virus-like particles (VLPs). With a dense and repetitive array of antigens displayed on the surface, VLPs can induce robust immune responses[24–28]. VLP vaccines against hepatitis B, human papillomavirus (HPV) and hepatitis E are among the most effective human vaccines, showing efficacies of 95–100% (ref. 28). The optimal antigen spacing has been determined using haptenated polymer molecules[29], with a minimum of 20–25 epitopes spaced by 5–10 nm deemed sufficient for effective B-cell activation. Recently, Schiller and Chackerian[30] elaborated the causes of why HIV-1 fails to rapidly induce neutralizing B-cell responses through a comparison of HIV-1 and HPV virions, which differ significantly in their surface antigen display. Self-assembling nanoparticles are of increasing interest to vaccine researchers, because they provide robust platforms to investigate the concept of particulate vaccines without involving complicated purification methods typically required for VLPs[31]. The 24-meric ferritin (FR) nanoparticle (12.2 nm in diameter) has been used to present the hemagglutinin (HA) of influenza[32,33], gp350 of Epstein–Barr virus[32] and scaffold antigens designed for HIV-1 and hepatitis C virus[34,35]. Recently, Sliepen et al.[36] reported the FR display of BG505 SOSIP trimer with immunogenicity tested in mice and rabbits. Two 60-meric nanoparticles—lumazine synthase (LS) from Aquifex aeolicus (14.8 nm in diameter) and dihydrolipoyl acetyltransferase (E2p) from Bacillus stearothermophilus (23.2 nm in diameter)—have also been reported in the design of multivalent HIV-1 immunogens. Specifically, LS was used as a carrier for an engineered gp120 outer domain (eOD) to target the germline precursors of VRC01-class bNAbs[37,38], while E2p was used to display the membrane-proximal external region (MPER) of gp41 (ref. 39), but neither antigen was presented in the native trimeric form. In principle, large nanoparticle platforms may be more advantageous for uptake by dendritic cells (DCs) and virus-like clustering of B-cell receptors (BCRs)[40–42].

Here we investigate the nanoparticle display of trimeric HIV-1 antigens by combining structural and antigenic analyses with B-cell activation assays. We first hypothesize that trimeric V1V2 and gp120 can be presented in native-like conformations around the threefold axes on the surface of nanoparticles. To test this hypothesis, we design constructs containing V1V2 and gp120 fused to the N terminus of FR subunit. These chimeric antigens can assemble into nanoparticles with high affinity for bNAbs targeting the apex, as well as other key epitopes, consistent with native-like trimer conformations. We then examine the particulate display of a stabilized gp140 trimer with a redesigned heptad repeat 1 (HR1) bend that shows significant improvement in trimer purity (described in the companion paper[43]). To facilitate this analysis, we design gp140-FR fusion constructs with different combinations of gp41 truncation and gp41-FR linker length. All gp140-FR nanoparticles bind to the apex-directed bNAbs with sub-picomolar affinities, with the MPER-containing gp140 nanoparticle also recognized by MPER-specific bNAb 4E10. The utility of 60-meric E2p to present trimeric gp120 and stabilized gp140 trimer in native-like conformations is also confirmed. Lastly, we demonstrate that the ExpiCHO expression system can markedly improve the yield and quality of a gp140-FR nanoparticle, and that three nanoparticles can trigger B cells carrying cognate VRC01 BCRs more effectively than gp140 trimer. We expect that the trimer-presenting nanoparticles designed and validated in this study will have important implications for HIV-1 vaccine development.

## Results

**FR nanoparticles presenting trimeric V1V2**. The V1V2 region of gp120 ranges from 50 to 90 residues in length with 1 in 10 residues N-glycosylated, forming a dense glycan shield on the HIV-1 Env[44]. The V1V2-encoded glycan epitopes can be recognized by bNAbs such as PG9 and PG16 (ref. 45), CH01-04 (ref. 46), PGT141-145 (ref. 47) and PGDM1400 (ref. 17). A short segment centered at N160 defines the specificity for most V1V2-directed bNAbs[48,49]. The crystal structure of scaffolded V1V2 in complex with PG9 has been determined for clade-C CAP45 and ZM109, revealing a Greek key motif with strands B and C harbouring two critical glycans[44,50]. The negative-stain electron microscopy (EM) reconstructions of PG9 and PGDM1400 in complex with the SOSIP trimer indicated that these two bNAbs are directed to the trimer apex with different angles of approach[12,17]. In this study, we hypothesized that the threefold axes on FR nanoparticle can be used to present V1V2 in the native-like trimeric conformation found in high-resolution structures of the SOSIP trimer[9,10] (Fig. 1a, left). We designed two constructs based on the V1V2 of clade-C ZM109: one containing all three disulfide bonds (residues 118–206, termed V1V2Ext) and a shortened version containing only two (residues 125–197, termed V1V2Sht), each fused to the N terminus (D5) of FR subunit[32,35] (Fig. 1a right, and Supplementary Table 1a). After fitting the C termini of trimeric V1V2 to the N termini of FR subunits around each threefold axis on the particle surface, structural modelling yielded Cα root-mean-square deviations (RMSD) of 3.7 and 0.8 Å for V1V2Ext-FR and V1V2Sht-FR, respectively (Fig. 1b). Further analysis revealed a dense glycan surface with diameters of 16.6 and 14.3 nm for V1V2Ext-FR and V1V2Sht-FR, respectively.

The two V1V2-FR constructs and the monomeric V1V2 were expressed transiently in N-acetylglucosaminyltransferase I-negative (GnTI[−/−]) HEK293S cells and purified using a Galanthus nivalis lectin (GNL) column followed by size exclusion chromatography (SEC) on a Superdex 200 10/300 GL column. The SEC profiles displayed a prominent peak indicative of well-formed nanoparticles (Fig. 1c), which were confirmed by blue native polyacrylamide gel electrophoresis (BN-PAGE) (Fig. 1d). Negative-stain EM revealed pure, homogeneous nanoparticles, as indicated by micrographs and two-dimensional (2D) class averages (Fig. 1e and Supplementary Fig. 1a). Purity and homogeneity appeared to be intrinsic to these V1V2-FR nanoparticles despite their differences in V1V2 length and the number of disulfide bonds contained. However, the V1V2 spikes were diffuse in the 2D class averages (Fig. 1e), indicative of some mobility. To probe the antigenicity of the V1V2 apex, we measured nanoparticle binding to PG9 (ref. 45), which

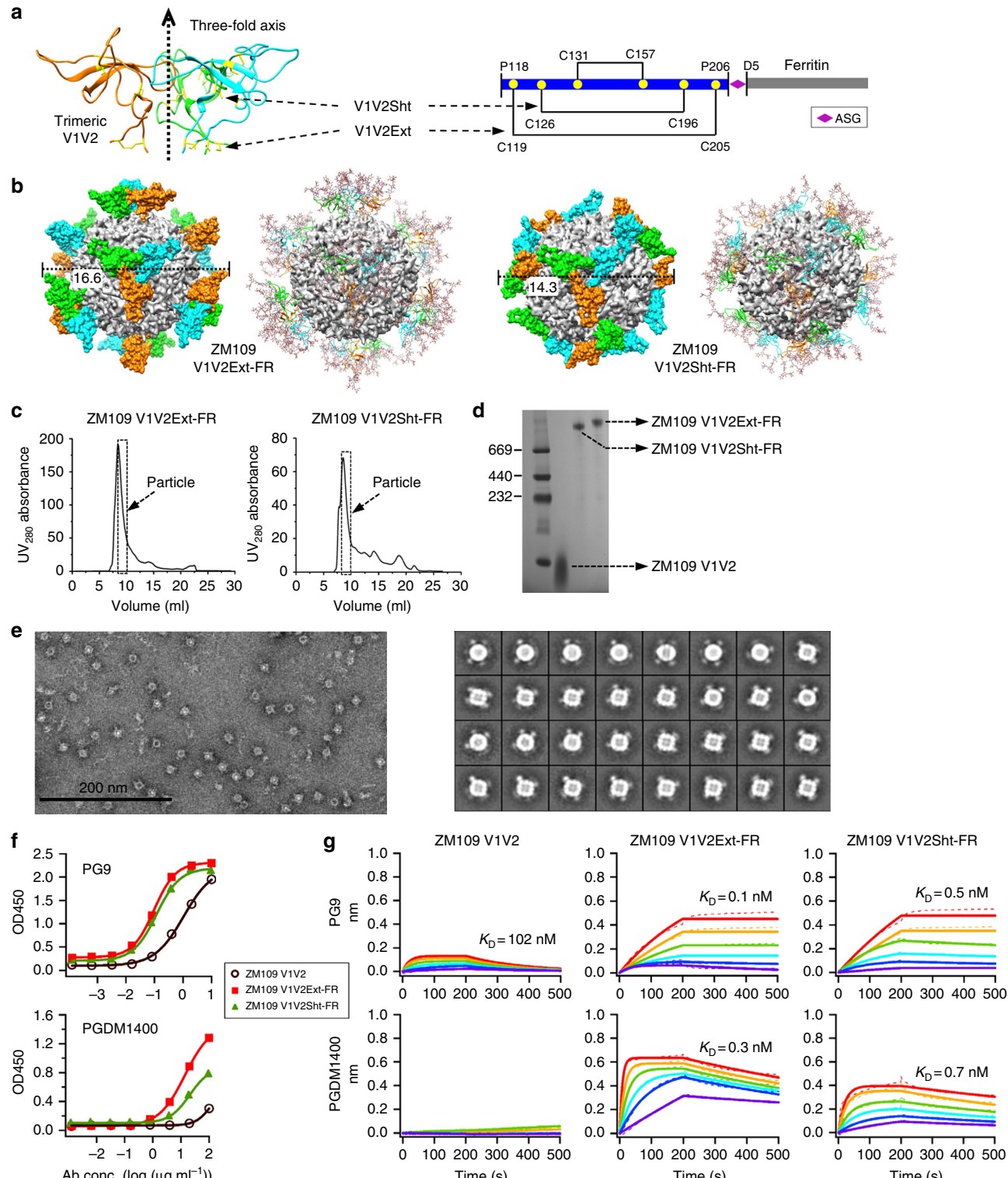

**Figure 1 | Design and characterization of V1V2-FR nanoparticles.** (**a**) Structure of trimeric V1V2 from clade-C ZM109 modelled on the BG505 SOSIP.664 trimer (PDB ID: 4TVP) (left) and schematic representations of V1V2Sht- and V1V2Ext-FR constructs (right). Three V1V2 chains are coloured in cyan, green and orange, respectively, with the disulfide bonds shown in yellow (side chains in stick model). (**b**) Structural models of V1V2Ext-FR (left) and V1V2Sht-FR (right) nanoparticles. For each design, the surface model of protein particle is shown on the left, whereas the ribbon model of trimeric V1V2 spikes decorated with *N*-linked glycans is shown on the right. FR and three V1V2 chains within each trimeric spike are coloured in grey, cyan, green and orange, respectively. (**c**) SEC profiles of V1V2Ext-FR (left) and V1V2Sht-FR (right) from a Superdex 200 10/300 GL column. Fractions used for EM and antigenic analyses (8.25–10.0 ml) are indicated with a dashed box. (**d**) BN-PAGE of monomeric V1V2 and two V1V2-FR nanoparticles. (**e**) Micrograph (left) and 2D class averages (right) of V1V2Ext-FR derived from negative-stain EM. (**f**) ELISA binding of monomeric V1V2 and two V1V2 nanoparticles to apex-directed bNAbs PG9 and PGDM1400. (**g**) Octet binding of monomeric V1V2 and two V1V2-FR nanoparticles to PG9 and PGDM1400. Sensorgrams were obtained from an Octet RED96 instrument using a titration series of six starting at the maximum of 500 nM for monomeric V1V2 and 50 nM for V1V2 nanoparticles, respectively. $K_D$ values are calculated from 1:1 global fitting for apex-directed bNAbs PG9 and PGDM1400.

recognizes V1V2 in both monomeric and trimeric forms[12,44,50], and PGDM1400, which only binds to the apex of the native-like SOSIP trimer[17]. Both V1V2-FR nanoparticles showed enhanced binding to PG9 and PGDM1400 in comparison with V1V2 monomer by enzyme-linked immunosorbent assay (ELISA) (Fig. 1f). Using biolayer interferometry (BLI) and immunoglobulin G (IgG), we characterized the kinetics of V1V2 binding to PG9 and PGDM1400 in both monomeric and particulate forms (Fig. 1g). As expected, V1V2 monomer bound to PG9 with low affinity and showed no binding to PGDM1400. By contrast, V1V2-FR nanoparticles bound to both bNAbs with sub-nanomolar affinities, but notably different kinetics. The faster on-rate observed for PGDM1400 indicated well-formed trimeric apexes that can be readily recognized by this bNAb. Of note, V1V2Sht-FR exhibited somewhat reduced affinities for both bNAbs in comparison with V1V2Ext-FR, suggesting an adversary effect of V1V2 shortening.

Our results demonstrate that the V1V2 region of HIV-1 Env can be presented in a native-like trimeric conformation on FR nanoparticle. We also observed homogenous nanoparticles for V1V2Ext-FR derived from clade-C CAP45 (Supplementary Fig. 1b–d), suggesting a design strategy adaptable to other strains. Overall, particulate display of trimeric V1V2 substantially improved its recognition by apex-directed bNAbs, supporting the notion that V1V2 nanoparticles may be promising immunogens to focus B-cell responses to this quaternary epitope.

**FR nanoparticles presenting trimeric gp120**. A 60-meric LS nanoparticle presenting an engineered gp120 core lacking variable loops (V1V2 and V3) and inner domain has been used to target the germline precursors of VRC01 (refs 37,38), a CD4-binding site (CD4bs)-directed bNAb[51]. However, structures of BG505 SOSIP trimer in complex with the VRC01-class bNAb PGV04 (ref. 52) revealed that glycans on the adjacent gp140 protomer are also involved in CD4bs recognition[9,10], suggesting an angle of approach constrained by the trimeric context. The importance of trimer constraints for HIV-1 neutralization was further demonstrated for human Ig knock-in mice, in which only BG505 SOSIP trimer, but not the LS-eOD nanoparticle, elicited NAb responses[53]. Based on these findings, we hypothesized that FR nanoparticle can be used to present full-length, trimeric gp120 and expose all its encoded bNAb epitopes in the SOSIP-like conformations. These nanoparticle designs may avoid the complications intrinsic to gp140 trimers containing the metastable gp41 and provide alternative 'trimer-like' immunogens. Here we designed three FR fusion constructs based on the gp120 of clade-A BG505: gp120Ext-FR and gp120Sht-FR differed in their gp120 length, while gp120SS-FR contained an additional disulfide bond aimed to stabilize the gp120 termini (Fig. 2a and Supplementary Table 1b). For gp120Ext-FR, structural modelling revealed a nearly perfect superposition of trimeric gp120 C termini (G495) and FR N termini (D5) around each threefold axis on the nanoparticle surface, with a Cα RMSD of 1.9 Å and a diameter of 26.2 nm (Fig. 2b).

The three gp120-FR constructs and the monomeric gp120 were expressed transiently in HEK293F cells. The secreted proteins were purified using a GNL column followed by SEC on a Superose 6 10/300 GL column. Among the three fusion constructs, gp120Sht-FR showed the most pronounced particle peak in the SEC profile, whereas gp120SS-FR appeared assembly deficient (Fig. 2c). Consistently, BN-PAGE exhibited high molecular weight (m.w.) bands for both gp120Ext-FR and gp120Sht-FR (Fig. 2d), while negative-stain EM revealed homogeneous nanoparticles with 2D class averages calculated for

gp120Sht-FR (Fig. 2e and Supplementary Fig. 2a,b). Overall, gp120Sht-FR displayed more efficient particle assembly and a surface decorated with well-formed gp120 spikes, as further confirmed by cryo-EM (Fig. 2e).

To assess the antigenicity of gp120-FR nanoparticles, we measured the kinetics of nanoparticle binding to a panel of representative bNAbs and non-NAbs (Fig. 2f). We first tested apex-directed bNAbs PG16 and PGDM1400. As expected, gp120 monomer exhibited minimal binding to PG16 and almost undetectable binding to PGDM1400. By contrast, two gp120 nanoparticles showed substantially enhanced binding to both bNAbs with sub-nanomolar affinities. This confirmed that three gp120s centered around each threefold axis on the FR nanoparticle can indeed form a SOSIP-like trimer conformation. For CD4bs-directed bNAb VRC01, both gp120Ext-FR and gp120Sht-FR displayed enhanced binding with flat dissociation curves. A similar trend was observed for NAb b12, which binds the CD4bs with a different angle of approach and footprint (Supplementary Fig. 2c). The avidity effect resulting from multivalent display was most pronounced for bNAb PGT121, which targets the V3 base and surrounding glycans: while monomeric gp120 bound weakly to PGT121 with a fast off-rate, gp120 nanoparticles showed strong binding with faster on-rates and flat dissociation curves. We then measured nanoparticle binding to non-NAbs. For CD4bs-directed F105, monomeric gp120 exhibited rapid on-/off-rates, whereas gp120-FR nanoparticles showed slow on-/off-rates. For V3-specific 19b, gp120 nanoparticles did show somewhat enhanced binding in comparison with monomeric gp120 (Supplementary Fig. 2c). For both F105 and 19b, the differences in kinetics observed between gp120 nanoparticles and monomer may be explained by the mixed effect of steric hindrance and avidity resulting from the dense display of gp120 trimers. Lastly, gp120 nanoparticles showed negligible binding to CD4i-specific 17b, in contrast to a notable recognition of gp120 monomer by this antibody (Supplementary Fig. 2c).

**60-meric nanoparticles presenting trimeric gp120**. Considering the previous reports of LS-eOD nanoparticles[37,38], we investigated whether such 60-mers could be used to present trimeric gp120. We selected two thermostable 60-mers with distinct structural features—LS[54] and E2p[55]—to examine this possibility. Compared with the 12.2-nm diameter of FR, LS is only slightly larger in size with a diameter of 14.8 nm (Supplementary Fig. 3a). Structural modelling indicated that the gp120Sht-LS nanoparticle surface would be covered entirely by 20 trimeric gp120 spikes with an estimated diameter of 28.7 nm (Supplementary Fig. 3a). Following transient expression in HEK293F cells and GNL purification, the secreted protein was analysed by SEC on a Superose 6 10/300 GL column. However, no particle peak was observed in the SEC profile of this LS fusion construct (Supplementary Fig. 3b). Consistently, BN-PAGE showed a predominant pentamer band, which was confirmed by negative-stain EM analysis of two SEC fractions (Supplementary Fig. 3c–e). Our results thus indicate that the LS nanoparticle platform may be suboptimal for displaying full-length gp120 trimers.

We next examined E2p, a hollow dodecahedron with a diameter of 23.2 nm[55] (Fig. 3a). Structural modelling yielded a gp120Sht-E2p nanoparticle with a diameter of 37.6 nm (Fig. 3a and Supplementary Table 1c), close to the optimal size for direct uptake by DCs[56]. The 293F-expressed, GNL-purified gp120Sht-E2p was analysed by SEC on a Superose 6 10/300 GL column, which showed a distinctive high m.w. peak corresponding to the chimeric E2p particles (Fig. 3b). Concordantly, BN-PAGE showed a concentrated nanoparticle

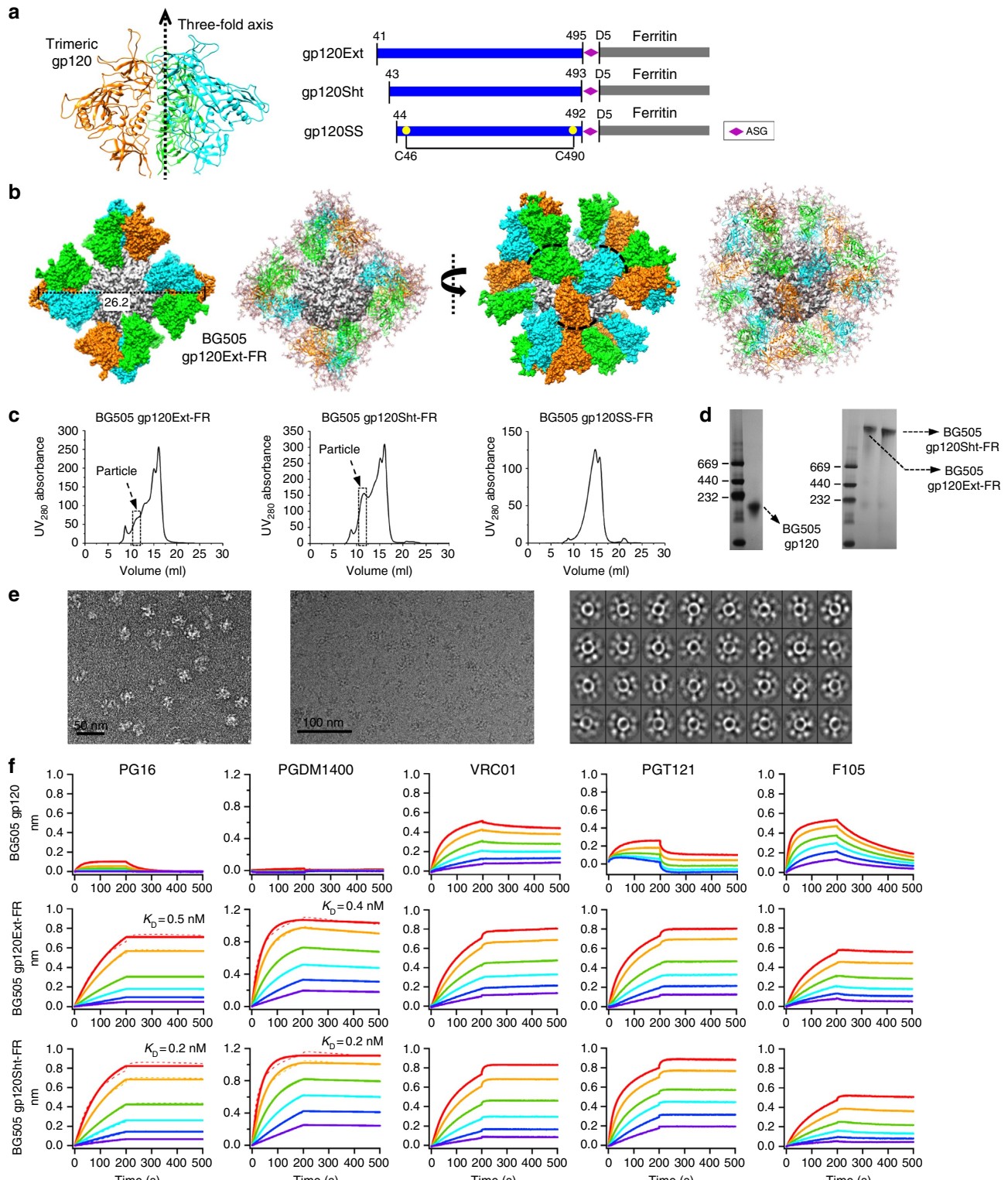

**Figure 2 | Design and characterization of gp120-FR nanoparticles.** (**a**) Structure of trimeric gp120 from clade-A BG505 modelled on the BG505 SOSIP.664 trimer (PDB ID: 4TVP) (left) and schematic representations of gp120Ext-, gp120Sht- and gp120SS-FR (FR) constructs (right). Three gp120 chains are coloured in cyan, green and orange, respectively. (**b**) Structural models of the gp120Ext-FR nanoparticle rotated by 90°. For each view, the surface model of protein particle is shown on the left, whereas the ribbon model of trimeric gp120 spikes decorated with *N*-linked glycans is shown on the right. FR and three gp120 chains within each trimeric spike are coloured in grey, cyan, green and orange, respectively. (**c**) SEC profiles of gp120Ext-FR (left), gp120Sht-FR (middle) and gp120SS-FR (right) from a Superose 6 10/300 GL column. Fractions used for EM and antigenic analyses (10.5–11.25 ml) are indicated with a dashed box. (**d**) BN-PAGE of monomeric gp120 and two gp120-FR nanoparticles. (**e**) EM analysis of gp120Sht-FR: micrograph derived from negative-stain EM (left), micrograph (middle) and 2D class averages (right) derived from cryo-EM. (**f**) Octet binding of monomeric gp120 and two gp120-FR nanoparticles to a panel of bNAbs and non-NAbs, with additional antibody binding data shown in Supplementary Fig. 2. Sensorgrams were obtained from an Octet RED96 instrument using a titration series of six starting at the maximum of 200 nM for monomeric gp120 and 50 nM for gp120 nanoparticles, respectively. $K_D$ values are calculated from 1:1 global fitting for apex-directed bNAbs PG16 and PGDM1400.

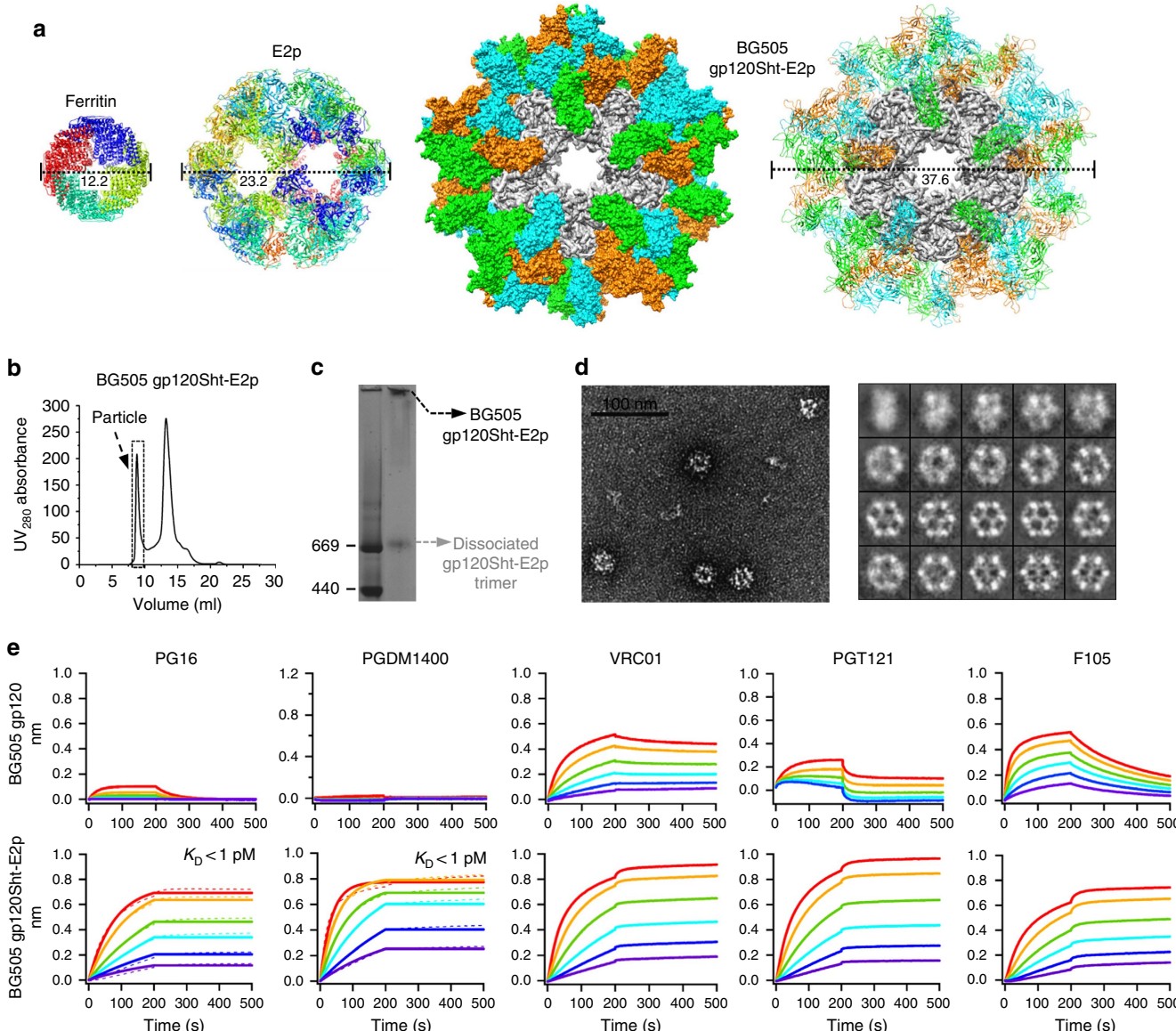

**Figure 3 | Design and characterization of 60-meric gp120-E2p nanoparticle.** (**a**) Structural models of FR, E2p and gp120Sht-E2p nanoparticles. The ribbon models of FR and E2p are colour-coded based on protein chains. For gp120Sht-E2p, the surface model of protein particle is shown on the left, whereas the ribbon model of trimeric gp120 spikes is shown on the right. E2p and three gp120 chains within each trimeric spike are coloured in grey, cyan, green and orange, respectively. (**b**) SEC profile of gp120Sht-E2p from a Superose 6 10/300 GL column. Fractions used for EM and antigenic analyses (8.5–9.75 ml) are indicated with a dashed box. (**c**) BN-PAGE of gp120Sht-E2p with both nanoparticle and potentially dissociated trimer bands labelled. (**d**) Micrograph (left) and 2D class averages (right) of gp120Sht-E2p derived from negative-stain EM. (**e**) Octet binding of monomeric gp120 and gp120Sht-E2p to a panel of bNAbs and non-NAbs, with additional antibody binding data shown in Supplementary Fig. 3. Sensorgrams were obtained from an Octet RED96 instrument using a titration series of six starting at the maximum of 15 nM for gp120Sht-E2p. $K_D$ values are calculated from 1:1 global fitting for apex-directed bNAbs PG16 and PGDM1400.

band on the gel, with a lighter band of low m.w. suggestive of dissociated gp120Sht-E2p trimer (Fig. 3c). Negative-stain EM revealed large, homogeneous nanoparticles (Fig. 3d, left), with 2D class averages showing hollow protein cages resembling the E2p crystal structure[55] (Fig. 3d right, and Fig. 3a). Unexpectedly, although negative-stain EM validated nanoparticle assembly, it also showed 2D class averages lacking the gp120 spikes. It was unclear from the EM analysis alone whether some, if not all, gp120s had formed trimeric spikes. To examine this issue, we measured nanoparticle binding to a panel of bNAbs and non-NAbs by BLI (Fig. 3e and Supplementary Fig. 3f). Remarkably, gp120Sht-E2p showed sub-picomolar affinities for apex-directed bNAbs PG16 and PGDM1400. The fast on-rates

and flat dissociation curves indicated native-like apexes resembling that of the SOSIP trimer, but with the additional advantage of avidity. Similar to gp120-FR nanoparticles, gp120Sht-E2p showed improved recognition by VRC01, PGT121 and b12. For non-NAbs, gp120Sht-E2p bound to CD4bs-specific F105 and V3-specific 19b at a similar level to gp120Sht-FR, but showed minimal binding to the CD4i antibody 17b.

As the receptor-binding protein, gp120 has been extensively studied as an HIV-1 vaccine candidate[57], but is now considered suboptimal due to the exposed non-neutralizing surface that should be buried within the native Env spike. Our results indicate that display of full-length gp120 on FR and E2p can restore

the native-like trimer conformation in the absence of gp41. With SOSIP-like antigenicity and variations in both size and surface spacing, these nanoparticles provide versatile platforms to reinvestigate gp120-based HIV-1 vaccines.

**FR nanoparticles presenting stabilized gp140 trimers.** The design and immunogenicity of a FR nanoparticle presenting BG505 SOSIP.664 trimer have been reported[36]. Compared with the SOSIP trimer, this gp140 nanoparticle showed notably reduced binding by ELISA to apex-directed bNAbs PG9 and PGT145, and to a bNAb targeting the gp120-gp41 interface, PGT151. This is somewhat surprising given the antigenic profiles we observed for the gp120 nanoparticles using BLI. Here we sought to approach the FR display of gp140 trimer with a redesigned HR1 bend (residues 547–569, termed HR1 redesign 1 (ref. 43)) and a detailed analysis of both linker length and gp41 truncation. The tested BG505 gp140 constructs included a gp140 truncated at position 664 (gp140.664), gp140.664 with a 10-residue linker (gp140.664-10aa) and a gp140 truncated at position 681 with a 10-residue linker (gp140.681-10aa) (Fig. 4a and Supplementary Table 1d). Structural modelling indicated diameters of 30.1, 35.7 and 40.1 nm for gp140.664-FR, gp140.664-10aa-FR and gp140.681-10aa-FR, respectively (Fig. 4b).

Since contaminant Env species cannot be eliminated during particle assembly, the purity of gp140 trimer will have a significant impact on the quality of resulting gp140 nanoparticles. To illustrate this problem, we compared the SEC profiles of the BG505 SOSIP.664 and HR1-redesigned trimers[43], which differed significantly in their purity (Fig. 4c). All gp140-FR fusion constructs, including a SOSIP-based design[36], were transiently co-expressed with furin in HEK293F cells and purified using GNL followed by SEC on a Superose 6 10/300 GL column. We first confirmed assembly of the reported SOSIP-FR nanoparticle[36] by SEC and negative-stain EM (Supplementary Fig. 4a,b). Our HR1-redesigned gp140-FR constructs exhibited similar SEC profiles, but without the misfolded monomer peak observed for SOSIP-FR (Supplementary Fig. 4c). Interestingly, negative-stain EM revealed an unknown protein species with a hexagonal structure mixed with aggregates and well-formed gp140 nanoparticles (Supplementary Fig. 4d). In this context, we examined the utility of Capto Core 700 column[58] for gp140 nanoparticle purification. Capto Core 700 and GNL columns together reduced non-particle impurities but not the hexagonal species, as indicated by SEC and negative-stain EM (Supplementary Fig. 4e,f). We next examined the combined use of Capto Core 700 and 2G12 affinity columns. With Env-specific 2G12 purification, gp140.664-10aa-FR showed the most visible particle peak in the SEC profile, in addition to a reduced trimer peak and no dimer/monomer peak (Fig. 4d). For all three HR1-redesigned gp140-FR constructs, BN-PAGE showed high m.w. bands corresponding to fully assembled nanoparticles (Fig. 4e). These BN bands are consistent with the SEC profiles (Fig. 4d) and the expected shift relative to the gp120-FR BN bands (Fig. 2d), but in contrast to the previous work showing SOSIP-FR nanoparticles at the top of the BN gel[36]. For gp140.664-10aa-FR, negative-stain EM revealed homogeneous nanoparticles with visible spikes displayed on the surface (Fig. 4f). The stability of this nanoparticle construct was confirmed by imaging a sample that had been frozen and thawed (Supplementary Fig. 4g). Fully assembled nanoparticles were also observed for gp140.664-FR and gp140.681-10aa-FR (Fig. 4g).

A large panel of bNAbs and non-NAbs was then used to characterize the antigenicity of gp140-FR nanoparticles with respect to the HR1-redesigned trimer[43] (Fig. 5 and Supplementary Fig. 5). We first used bNAbs PGDM1400, PGT145 and PG16 to probe the apex of gp140 trimers

displayed on the nanoparticle surface. Remarkably, all gp140-FR nanoparticles showed sub-picomolar binding affinities with flat dissociation curves due to avidity. For bNAb PGT121, which targets the N332 supersite, gp140 nanoparticles showed similar binding profiles to that of the HR1-redesigned trimer with an increased on-rate. For CD4bs-directed bNAb VRC01, gp140-FR nanoparticles displayed slow association similar to gp120-FR nanoparticles. For bNAbs PGT151 and 35O22, which target the gp120–gp41 interface, we observed different kinetics. For PGT151 (refs 13,14), a faster association phase was observed with an unchanged dissociation pattern. By contrast, 35O22 (ref. 59) showed an elevated on-rate that was accompanied by an increased off-rate. Overall, all three gp140-FR nanoparticles showed improved recognition by bNAbs, except for VRC01, suggesting that the crowded surface display of gp140 trimers may affect this class of bNAbs[60,61]. For non-NAbs, decreased binding was observed for CD4bs-specific F105 and b6, with a more significant reduction for b6. For V3-specific 19b, a slower association phase was observed for gp140 nanoparticles compared with the individual trimer and gp120 nanoparticles (Supplementary Figs 2c and 3f), suggesting a minimized V3 exposure due to the dense display of gp140 trimers. For F240, which targets an immunodominant epitope in cluster I of gp41, gp140 nanoparticles exhibited undetectable binding compared to the residual binding observed for individual trimer. The gp140 nanoparticles showed no detectable binding to CD4i-specific 17b and A32, similar to the individual trimer. Lastly, we used bNAbs 4E10 and 10E8 to probe the antigenicity of MPER in the context of gp140.681-10aa-FR (Fig. 5c). Although this nanoparticle exhibited strong binding to 4E10 with rapid association and flat dissociation curves, it bound only weakly to 10E8. Structural modelling revealed that MPER is proximal to the FR particle surface with a distance of more than 10 nm from the outer glycan surface (Fig. 4b, right), suggesting that steric hindrance may have a significant impact on 10E8, which selects for a conformational epitope extending beyond the 4E10-binding site[16].

Our analyses thus revealed salient features of gp140-FR nanoparticles with a redesigned HR1 bend[43]. In contrast to the previous report[36], antigenic profiling by BLI clearly demonstrated enhanced nanoparticle binding to bNAbs and reduced binding to non-NAbs. The intrinsic purity of the HR1-redesigned trimer[43] proved essential for the production of FR nanoparticles with homogeneous, native-like Env spikes. Taken together, our results suggest that these gp140 nanoparticles may be superior to gp140 trimers in eliciting robust B-cell responses to bNAb epitopes.

**60-meric nanoparticles presenting stabilized gp140 trimer.** We examined the utility of LS and E2p 60-mers to display the HR1-redesigned BG505 gp140 trimer[43]. Given the high density of display sites on LS[54], we first designed a construct containing a 10-residue linker between the C terminus of LS subunit and the N terminus of gp140 (Supplementary Table 1e). Structural modelling yielded an estimated diameter of 39.2 nm for LS-10aa-gp140.664 (Supplementary Fig. 6a). Following furin co-expression in HEK293F cells and GNL purification, the secreted protein was analysed by SEC and negative-stain EM (Supplementary Fig. 6b,c), in which nanoparticles were not identified. We next designed an E2p fusion construct by connecting the C terminus of gp140 to the N terminus of E2p core subunit (Supplementary Table 1e). Structural modelling indicated that the gp140.664-E2p nanoparticle, 41.5 nm in diameter, can expose all bNAb epitopes on the Env trimer (Fig. 6a). The designed gp140-E2p construct was co-transfected with furin in HEK293F cells followed by GNL purification and SEC on a Superose 6 10/300 GL column. Although the overall

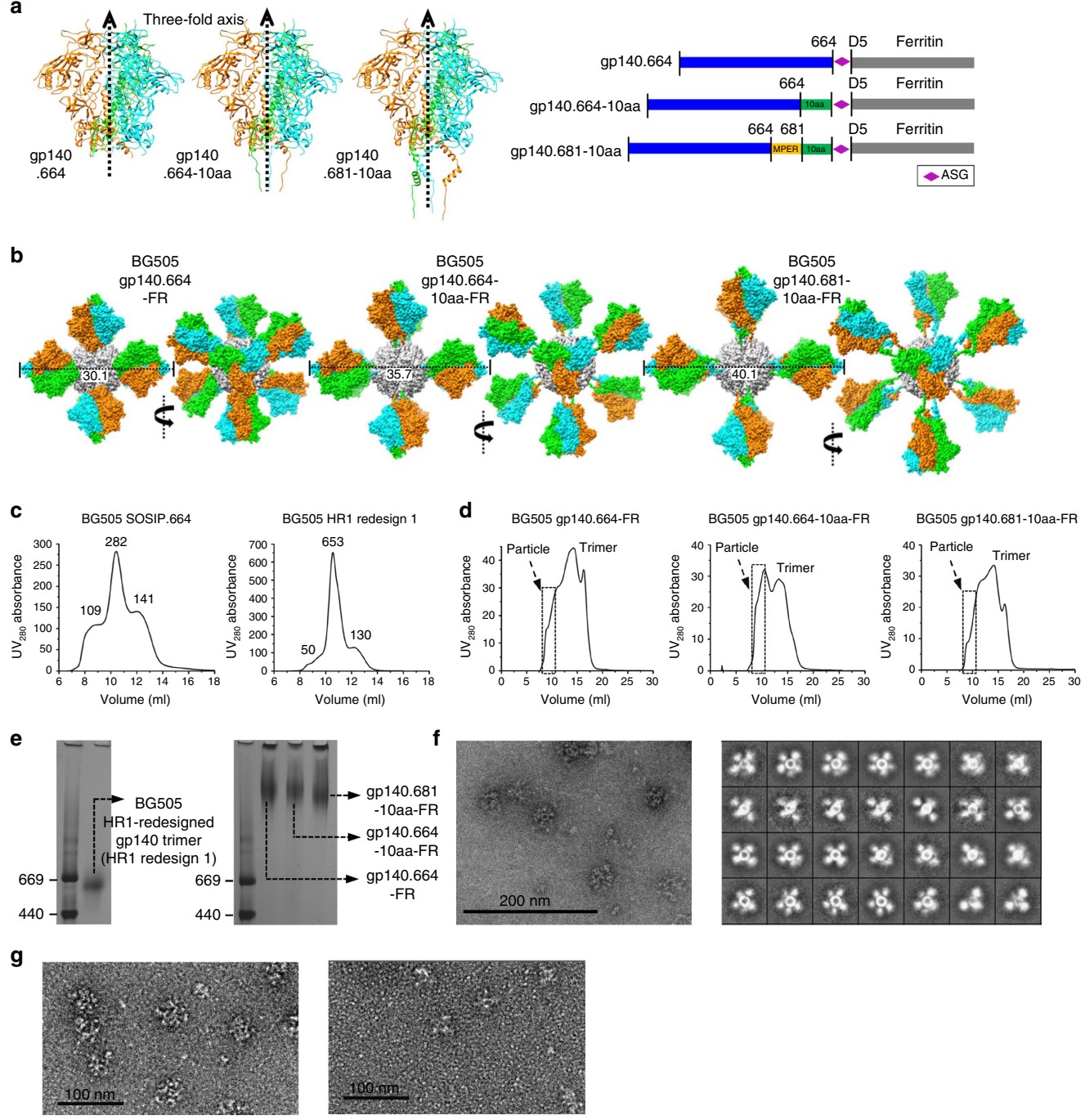

**Figure 4 | Design and structural characterization of gp140-FR nanoparticles.** (**a**) Structures of three BG505 gp140 trimers modelled on the BG505 SOSIP.664 trimer (PDB ID: 4TVP) (left) and schematic representations of gp140.664-, gp140.664-10aa- and gp140.681-10aa-FR constructs (right), all containing a redesigned HR1 bend. Three gp140 chains are coloured in cyan, green and orange, respectively. (**b**) Surface models of gp140.664-FR, gp140.664-10aa-FR and gp140.681-10aa-FR nanoparticles, each presenting two views rotated by 90°. FR and three gp140 chains within each Env trimer are coloured in grey, cyan, green and orange, respectively. (**c**) SEC profiles of SOSIP.664 (left) and HR1-redesigned (right) gp140 trimers from a Superdex 200 10/300 GL column. The absolute ultraviolet absorption at 280 nm (UV$_{280}$) absorbance values of aggregate (at 9 ml), trimer (at 10.5 ml) and dimer/monomer (at 12 ml) peaks are labelled for comparison. (**d**) SEC profiles of gp140.664-FR (left), gp140.664-10aa-FR (middle) and gp140.681-10aa-FR (right) from a Superose 6 10/300 GL column after purification using Capto Core 700 and 2G12 affinity columns. Fractions used for EM and antigenic analyses (8.0–10.75 ml) are indicated with a dashed box. (**e**) BN-PAGE of HR1-redesigned gp140 trimer and three gp140-FR nanoparticles. (**f**) Micrograph (left) and 2D class averages (right) of gp140.664-10aa-FR derived from negative-stain EM. (**g**) Micrographs of gp140.664-FR (left) and gp140.681-10aa-FR (right) derived from negative-stain EM.

expression was low, the most prominent peak in the SEC profile corresponded to well-formed gp140-E2p nanoparticles (Fig. 6b), which were confirmed by a high m.w. band at the top of the BN gel (Fig. 6c). Consistently, homogeneous nanoparticles with a dense layer of spikes were observed by negative-stain EM (Fig. 6d). Together, our results indicate that gp140.664-E2p can form VLP-size nanoparticles displaying a regular array of Env trimers on the surface poised to engage immune responses.

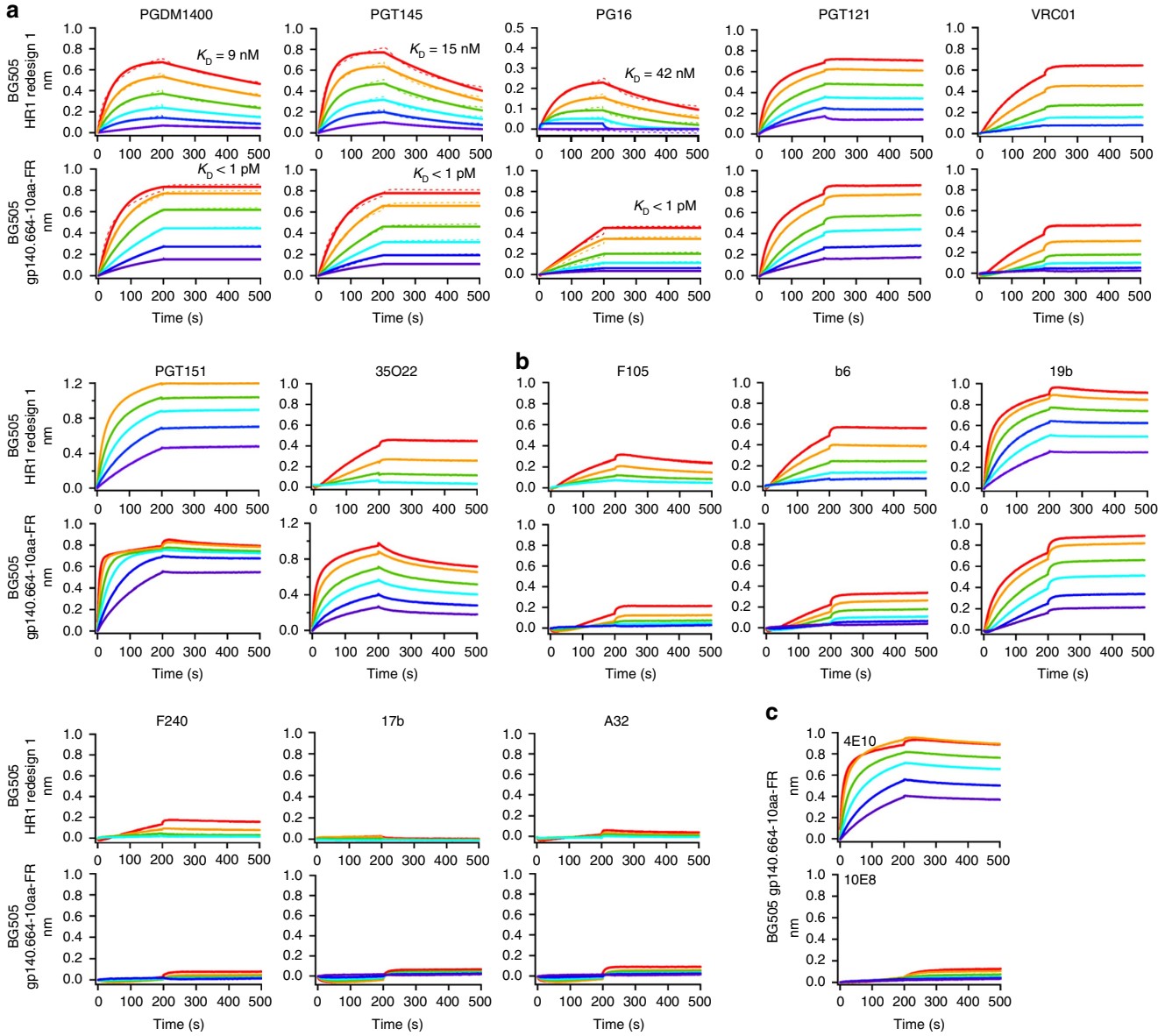

**Figure 5 | Antigenic evaluation of gp140-FR nanoparticles.** Octet binding of HR1-redesigned gp140 trimer and gp140.664-10aa-FR nanoparticle to a panel of (**a**) bNAbs and (**b**) non-NAbs. (**c**) Octet binding of gp140.681-10aa-FR to MPER-directed bNAbs 4E10 and 10E8. For gp140.664-FR and gp140.681-10aa-FR, detailed antibody binding profiles are shown in Supplementary Fig. 5. Sensorgrams were obtained from an Octet RED96 using a titration series of six starting at the maximum of 200 nM for HR1-redesigned gp140 trimer and 35 nM for gp140.664-10aa-FR nanoparticle. $K_D$ values are calculated from 1:1 global fitting for apex-directed bNAbs PGDM1400, PGT145 and PG16 in (**a**).

We first characterized the gp140.664-E2p nanoparticle using a panel of bNAbs (Fig. 6e). For apex-directed bNAbs PG16 and PGDM1400, gp140-E2p exhibited slow association curves, flat dissociation curves and sub-picomolar affinities. For CD4bs-directed bNAb VRC01, gp140-E2p showed relatively low binding and flat dissociation curves reminiscent of the gp140-FR nanoparticles (Fig. 5a). For bNAb PGT121, we observed native-like trimer kinetics with slightly reduced on-rate. For bNAbs PGT151 and 35O22, gp140-E2p exhibited binding profiles similar to gp140-FR nanoparticles. Overall, gp140.664-E2p showed reduced recognition by bNAbs relative to the individual trimer and gp140-FR nanoparticles, with the most visible change observed for VRC01 and the least for PGT151. This raised the possibility that non-native gp140 trimers were displayed on the E2p surface. To examine this possibility, we assessed nanoparticle binding to a panel of non-NAbs by BLI (Fig. 6f). For CD4bs-specific F105, we observed weakened binding

with a faster off-rate relative to the individual trimer. Remarkably, the 19b binding profile revealed a significant reduction in V3 exposure, in contrast to the enhancement observed for all other gp120 and gp140 nanoparticles. The gp140-E2p nanoparticle also bound to the CD4i antibody 17b at a minimal level. However, gp140-E2p exhibited somewhat increased recognition by gp41-specific F240 relative to the individual trimer and gp140-FR nanoparticles. Further analysis of the gp140 fusion sites on both E2p and FR nanoparticles (Supplementary Fig. 7) suggested that the increased F240 binding could be the result of structural strain in the gp41-E2p connecting region that may require further optimization.

The gp140.664-E2p nanoparticle, with an optimal size (40–50 nm) for DC uptake[56], may be more advantageous than gp140-FR nanoparticles in eliciting robust immune responses. Based on our observations of gp140.681-10aa-FR, it is possible that the HR1-redesigned gp140.681 trimer can also be displayed

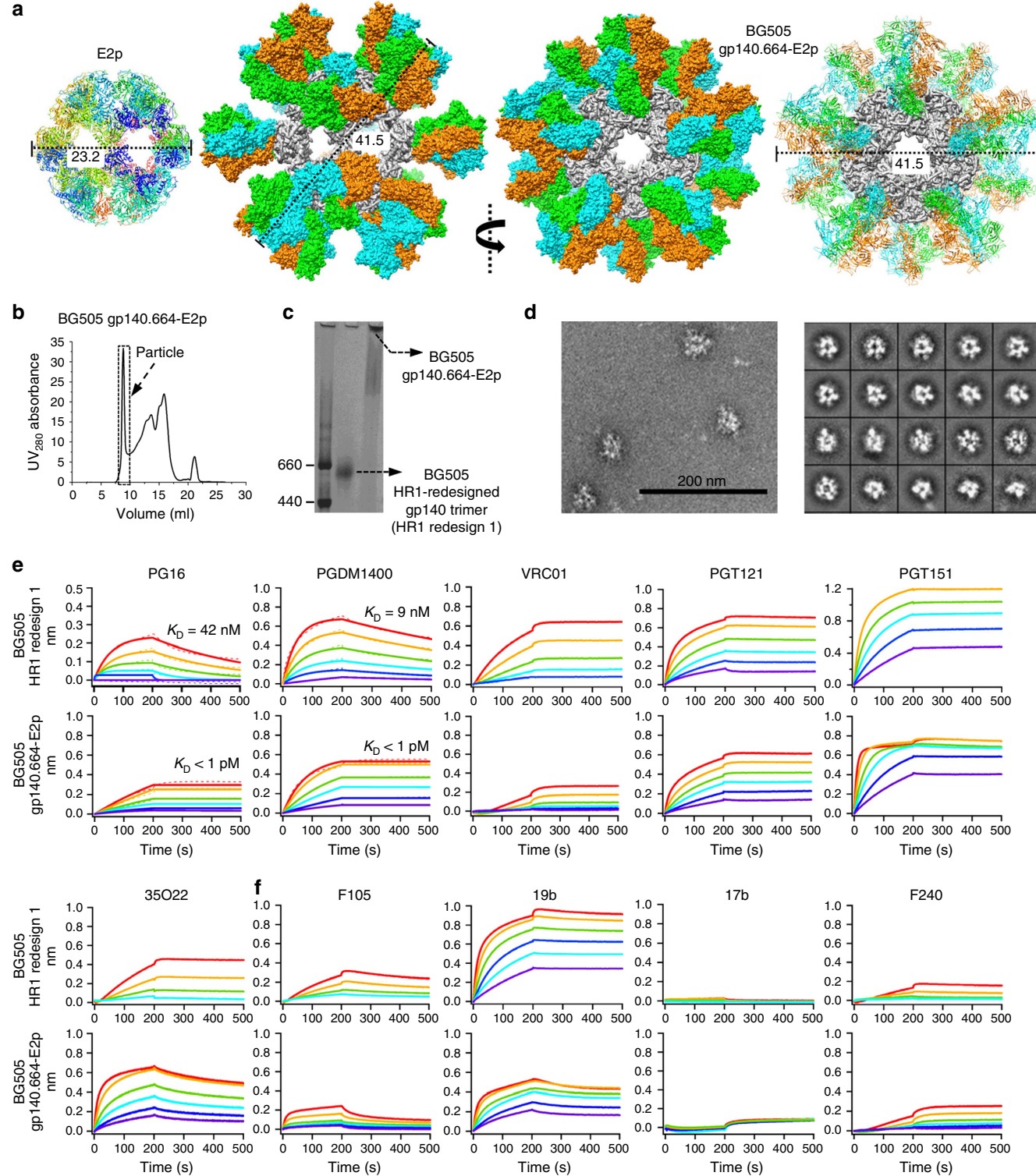

**Figure 6 | Design and characterization of 60-meric gp140-E2p nanoparticle.** (**a**) Structural models of E2p and gp140.664-E2p nanoparticle. The ribbon model of E2p is colour-coded by protein chains (left). The surface model of gp140.664-E2p is presented in two different views, with the second review centred around a fivefold axis (middle), for which the gp140 trimers are also depicted as ribbons (right). E2p and three gp140 chains within each Env trimer are coloured in grey, cyan, green and orange, respectively. (**b**) SEC profile of gp140.664-E2p from a Superose 6 10/300 GL column. Fractions used for EM and antigenic analyses (8.0–10.0 ml) are indicated with a dashed box. (**c**) BN-PAGE of HR1-redesigned gp140 trimer and gp140.664-E2p nanoparticle. (**d**) Micrograph (left) and 2D class averages (right) of gp140.664-E2p derived from negative-stain EM. Octet binding of HR1-redesigned gp140 trimer and gp140.664-E2p nanoparticle to a panel of (**e**) bNAbs and (**f**) non-NAbs, with additional antibody binding data shown in Supplementary Fig. 6. Sensorgrams were obtained from an Octet RED96 instrument using a titration series of six starting at a maximum of 16 nM for gp140.664-E2p. $K_D$ values are calculated from 1:1 global fitting for apex-directed bNAbs PG16 and PGDM1400 in **e**.

on E2p in a similar manner. In summary, E2p provides a versatile nanoparticle platform for the development of trimer-presenting, VLP-like HIV-1 vaccines.

**High-yield production and effective B-cell stimulation.** Robust nanoparticle expression is important for industrial-scale production and immunogenicity studies. In this context, we examined the utility of the ExpiCHO transient expression system, using gp140.664-10aa-FR as a test case. The secreted protein was harvested 2 weeks after transfection and purified using a 2G12 affinity column followed by SEC on a Superose 6 10/300 GL column. The 200-ml ExpiCHO expression produced a particle peak with a ultraviolet 280 nm absorbance value that is twofold higher than that from the 2-l HEK293F expression (Fig. 7a). A small particle peak was still visible in the SEC profile of the 2G12 flow-through after additional GNL purification. Overall, we estimated a ≥20-fold increase in particle yield. Negative-stain EM confirmed the structural homogeneity of this ExpiCHO-produced nanoparticle (Fig. 7b), which also demonstrated improved antigenicity in comparison with the HEK293F-produced nanoparticle (Fig. 7c), with significantly enhanced binding to the apex-directed bNAb PGDM1400.

Avidity differentiates VLPs and nanoparticles from individual antigens by their B-cell responses *in vivo*. Thus, surface plasmon resonance (SPR) was used to assess whether IgG can cross-react with two gp140 trimers on the nanoparticle surface for bNAbs PGDM1400 and PGT151 (Table 1). For gp140.664-E2p, faster on-rates and slower off-rates were observed for IgGs relative to their cognate Fabs, indicating interactions between one IgG and two adjacent trimers on the E2p surface. By contrast, the differences in kinetics between IgGs and Fabs were negligible for gp140.664-10aa-FR, suggesting a less favourable IgG cross-reaction on the FR surface perhaps due to its small size and greater curvature. We also investigated the ability of gp120 and gp140 nanoparticles to stimulate B cells carrying cognate receptors that can recognize bNAb epitopes on the Env. To this end, we stimulated a B-cell line carrying the VRC01 BCR[62] either with antibodies directed to the VRC01 light chain (anti-human κ-chain) as a positive control or with HIV-1 antigens. Nanoparticles gp120Sht-FR, gp120Sht-E2p and gp140.664-10aa-FR were compared against the HR1-redesigned gp140 trimer. Robust B-cell triggering was observed for the three nanoparticles presenting multiple copies of gp120 or gp140; whereas the individual gp140 trimer displayed a notably weaker response (Fig. 7d). Our results thus indicate that these nanoparticles may effectively engage B cells bearing bNAb characteristics *in vivo* and suggest that a diverse array of nanoparticle constructs reported here is capable of virus-like BCR clustering. Future studies focusing on B-cell stimulation and immunogenicity will facilitate the rational comparison and selection of these particulate HIV-1 immunogens (Fig. 7e).

## Discussion

HIV-1 vaccine development is entering a new era as high-resolution Env structures and panels of bNAbs become available[63]. In particular, the BG505 SOSIP.664 trimer has established a rigorous foundation for rational vaccine design[6]. Given the current focus on gp140 trimer design in HIV-1 research[18–23], the reported immunogenicity of the SOSIP trimer will provide a baseline for comparison with forthcoming trimer-based vaccines[21]. However, the lower-than-expected trimer immunogenicity[21] suggests that factors beyond trimer stabilization have to be taken into account in future vaccine design. One such critical factor is the multivalent display of native Env spikes[30]. Although the use of HIV-1 virions bearing various forms of Env has been explored[64,65], sparse display and undetermined spacing of trimeric spikes may not be optimal for eliciting robust B-cell responses[30]. Thus, self-assembling nanoparticles with proper size and surface spacing that mimic VLPs would be of high priority in future vaccine development. Recently, Sliepen *et al.* reported the design and immunogenicity of a FR nanoparticle presenting the BG505 SOSIP.664 trimer[36]. However, the suboptimal structural and antigenic properties of this nanoparticle highlighted the need for a more in-depth analysis.

In this study, we sought to extend the limit of multivalent HIV-1 vaccine design with a structure-based, hypothesis-driven approach. Based on the SOSIP trimer structures[9,10,11], we assessed particulate display of trimeric Env antigens on FR, LS and E2p nanoparticle platforms. We showed for the first time that V1V2 and gp120 can form native-like trimeric spikes on FR and E2p nanoparticles. With simple design, efficient production and SOSIP-like antigenic profiles, these nanoparticles provide useful tools to investigate B-cell responses to the apex and other bNAb epitopes in a multivalent form. By contrast, the inherent Env metastability has posed a great challenge to the particulate display of gp140 trimer. Although bNAb affinity purification is effective for individual gp140 trimer[18], it may not be optimal for gp140 nanoparticles, as gp140 trimerization and particle assembly will likely occur simultaneously. Therefore, the intrinsic purity of gp140 trimer is critical to nanoparticle vaccine design using the gene fusion strategy. To address this issue, we adopted a novel gp140 trimer with a redesigned HR1 bend that significantly improved the purity of well-folded trimers[43]. When this trimer was displayed on FR and E2p, the resulting nanoparticles produced desirable antigenic profiles against a panel of bNAbs and non-NAbs. Of note, an alternative approach to gene fusion is to chemically conjugate the purified gp140 trimers to the surface of a nanoparticle or VLP, as evidenced by the development of bacteriophage $Q_\beta$-based VLP vaccines[25].

Our study also demonstrated how size and surface spacing affect the ability of a nanoparticle to display large, complex HIV-1 antigens. The 60-meric LS (14.8 nm) is only marginally larger than the 24-meric FR (12.2 nm), but significantly smaller than the 60-meric E2p (23.2 nm). Although LS has been successfully used to display a truncated version of gp120 (refs 37,38), the LS fusion constructs containing full-length gp120 and gp140 were unable to assemble, probably due to insufficient spacing between the displayed trimeric antigens. The results for FR and E2p demonstrated that greater surface spacing may allow for more efficient nanoparticle assembly and antibody access to the epitopes near the nanoparticle surface. However, over-distanced antigens on the particle surface may increase flexibility and lead to less effective cross-linking of cognate BCRs *in vivo*. Nevertheless, robust triggering of cognate VRC01 receptors in B-cell activation assays confirmed that appropriate surface spacing enabled effective BCR clustering by gp120 or gp140 trimers presented on FR and E2p nanoparticles.

In summary, the rational design, structural analysis, antigenic profiling and B-cell activation assays in this study have established a novel array of trimer-presenting nanoparticles with potential as HIV-1 vaccine antigens that merit detailed investigation of immunogenicity.

## Methods

**Nanoparticle design and modelling.** A Perl script was developed to (1) identify threefold display sites on the surface of a given nanoparticle; (2) superpose the C termini of a trimeric HIV-1 antigen onto the N termini of three particle subunits around each threefold axis on the nanoparticle surface; and (3) generate coordinates for the resulting trimer-presenting nanoparticle with various structural parameters calculated at the completion of the process. The nanoparticle model was then visualized using UCSF Chimera (https://www.cgl.ucsf.edu/chimera/) for manual inspection and determination of the antigen-particle linker length. The computer code for nanoparticle modelling can be obtained from J.Z. upon request.

**Antibodies.** We used a panel of bNAbs and non-NAbs to characterize the antigenicity of designed trimer-presenting nanoparticles. 2G12 and b12, as well as non-

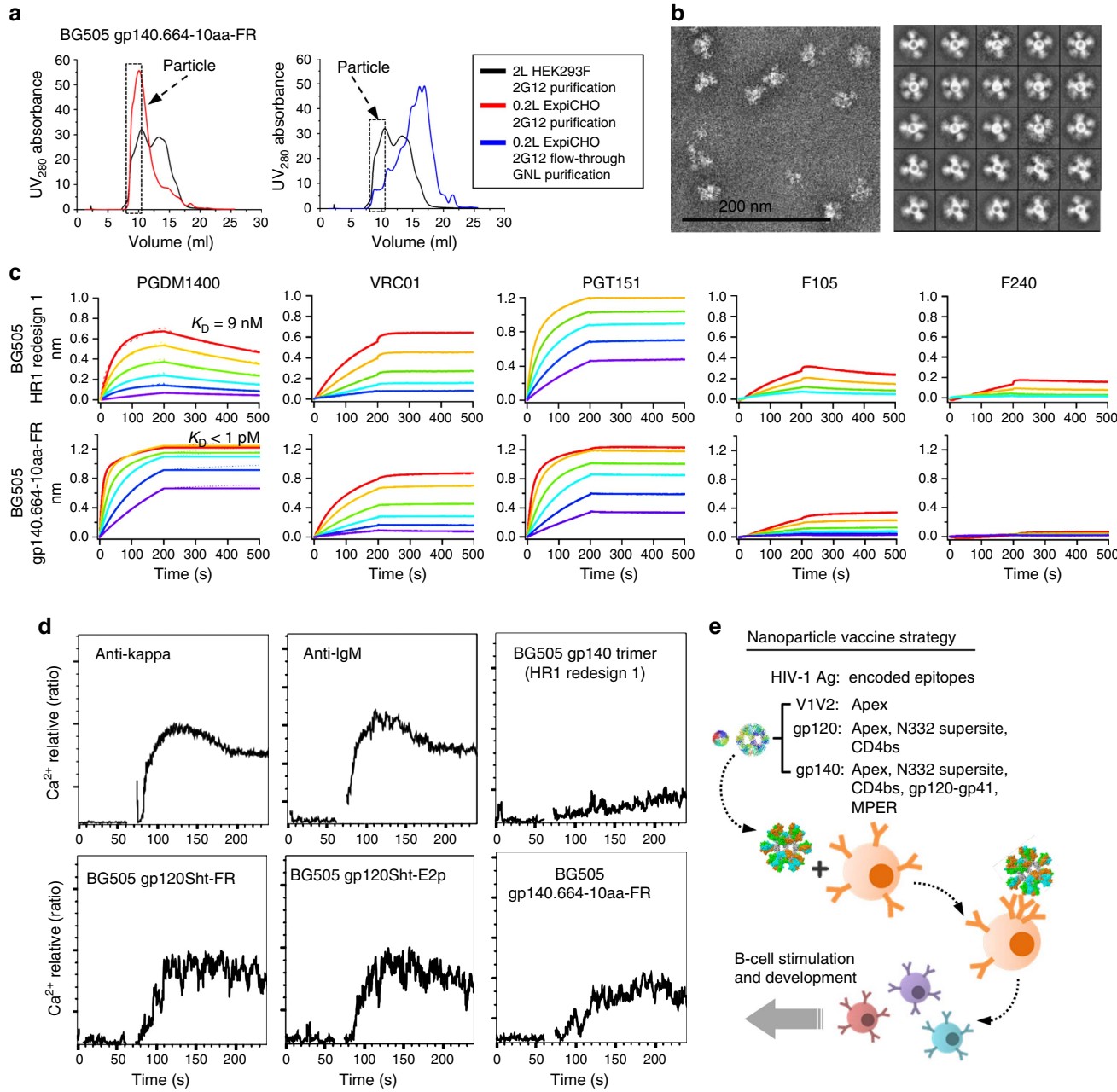

**Figure 7 | High-yield nanoparticle expression and effective B-cell stimulation.** (**a**) SEC profiles of 200-ml ExpiCHO-produced gp140.664-10aa-FR following 2G12 purification (red line, left) and 2G12 flow-through after additional GNL purification (blue line, right). The SEC profile of 2-liter HEK293F-produced gp140.664-10aa-FR after 2G12 purification (black line) is included for comparison. All SEC profiles were obtained from a Superose 6 10/ 300 GL column. Fractions used for EM and antigenic analyses (8.0–10.75 ml) are indicated with a dashed box. (**b**) Micrograph (left) and 2D class averages (right) of ExpiCHO-produced gp140.664-10aa-FR derived from negative-stain EM. (**c**) Octet binding of HR1-redesigned gp140 trimer and ExpiCHO-produced gp140.664-10aa-FR to three bNAbs and two non-NAbs. Sensorgrams were obtained from an Octet RED96 instrument using a titration series of six starting at a maximum of 16 nM for gp140.664-E2p. $K_D$ values are calculated from 1:1 global fitting for the apex-directed bNAb PGDM1400. (**d**) $Ca^{2+}$ mobilization in B-cell transfectants carrying VRC01 BCR. WEHI231 cells expressing a doxycyclin-inducible form of VRC01 BCR were stimulated with anti-BCR antibodies or the indicated antigens at a concentration of 10 μg ml$^{-1}$: anti-human Ig κ-chain F(ab′)$_2$; anti-mouse IgM; HR1-redesigned gp140 trimer, gp120Sht-FR, gp120Sht-E2p or gp140.664-10aa-FR. (**e**) Schematic view of the nanoparticle vaccine strategy. Structures of FR and E2p nanoparticles are shown with three trimeric antigens and their encoded bNAb epitopes. Nanoparticle-induced BCR clustering and B-cell lineage maturation leading to bNAbs are depicted schematically.

NAbs F240, 7B2, 17b and A32 were requested from the NIH AIDS Reagent Program (https://www.aidsreagent.org/). Other bNAbs and non-NAbs were provided by D.S. and D.R.B. Fabs used in the SPR analysis were provided by B.Z.

**HIV-1 antigen expression and purification.** V1V2 monomer and nanoparticles were transiently expressed in *N*-acetylglucosaminyltransferase I-negative (GnTI$^{-/-}$) HEK293S cells, whereas all other HIV-1 antigens (gp120 monomer,

gp140 trimer and gp120/gp140 nanoparticles) were transiently expressed in HEK293F cells (Life Technologies, CA). Briefly, HEK293 F/S cells were thawed and incubated with FreeStyle 293 Expression Medium (Life Technologies) in a Shaker incubator at 37 °C, with 120 r.p.m. and 8% $CO_2$. When the cells reached a density of $2.0 \times 10^6$ ml$^{-1}$, expression medium was added to reduce cell density to $1.0 \times 10^6$ ml$^{-1}$ for transfection with polyethyleneimine (PEI) (Polysciences, Inc). For V1V2- and gp120-based antigens, 900 μg of antigen plasmid was added to 25 ml of Opti-MEM transfection medium (Life Technologies) and then mixed with 5 ml of

**Table 1 | SPR kinetic analysis of BG505 gp140 nanoparticles binding to bNAbs in IgG and Fab forms.***

| Nanoparticle | bNAb | $k_a$ (M$^{-1}$s$^{-1}$) | | $k_d$ (s$^{-1}$) | | $K_D$ (M) | |
|---|---|---|---|---|---|---|---|
| | | IgG | Fab | IgG | Fab | IgG | Fab |
| gp140.664-10aa-FR | PGDM1400 | $1.12 \times 10^5$ | $1.38 \times 10^5$ | $4.74 \times 10^{-4}$ | $6.62 \times 10^{-4}$ | $4.22 \times 10^{-9}$ | $4.80 \times 10^{-9}$ |
| | PGT151 | $1.75 \times 10^5$ | $1.20 \times 10^5$ | $9.86 \times 10^{-4}$ | $7.08 \times 10^{-4}$ | $5.63 \times 10^{-9}$ | $5.88 \times 10^{-9}$ |
| gp140.664-E2p | PGDM1400 | $2.58 \times 10^5$ | $6.45 \times 10^4$ | $5.49 \times 10^{-4}$ | $1.95 \times 10^{-3}$ | $2.13 \times 10^{-9}$ | $3.03 \times 10^{-8}$ |
| | PGT151 | $4.27 \times 10^5$ | $1.28 \times 10^5$ | $5.94 \times 10^{-4}$ | $1.00 \times 10^{-3}$ | $1.29 \times 10^{-9}$ | $7.85 \times 10^{-9}$ |

SPR, surface plasmon resonance.
*SPR analysis of gp140 nanoparticle–bNAb binding kinetics was performed on a Biacore 3000 instrument using a custom wizard application (see details in Methods).

PEI–MAX (1.0 mg ml$^{-1}$) in 25 ml of Opti-MEM, whereas for gp140-based antigens, 800 µg of antigen plasmid and 300 µg of furin plasmid were mixed for transfection. After incubation for 30 min, the DNA–PEI–MAX complex was added to 1-litre 293F/S cells. Culture supernatants were harvested 5 days after transfection, clarified by centrifugation at 1,800 r.p.m. for 22 min and filtered using 0.45-µm filters (Thermo Fisher Scientific). HIV-1 antigen proteins were extracted from the supernatant using a GNL column (Vector Labs). The bound proteins were eluted with PBS containing 500 mM NaCl and 1 M methyl-α- D-mannopyranoside and purified by SEC on a Superdex 200 Increase 10/300 GL column or a Superose 6 10/300 GL column (GE Healthcare). Protein concentration was determined using ultraviolet absorption at 280 nm (UV$_{280}$) with theoretical extinction coefficients. For gp140-FR nanoparticles, Capto 700 Core column (GE Healthcare) and 2G12 affinity column[6] were used in various combinations with SEC to improve the particle purity. The ExpiCHO transient expression system (Thermo Fisher Scientific) was used to improve the particle yield for the gp140.664-10aa-FR construct. ExpiCHO cells and expression media were purchased from the vendor (Thermo Fisher Scientific), with the transfection performed according to the vendor's protocol. Protein was harvested from the supernatant 2 weeks after co-transfection with furin and subjected to Env-specific purification using a 2G12 affinity column.

**BN-PAGE.** Individual antigens and trimer-presenting nanoparticles were analysed by BN-PAGE and stained using Coomassie blue. The protein samples were mixed with G250 loading dye and added to a 4–12% Bis-Tris NuPAGE gel (Life Technologies). BN-PAGE gels were run for 2 h at 150 V using NativePAGE running buffer (Life Technologies) according to the manufacturer's instructions.

**Negative-stain EM.** The purified nanoparticles were analysed by negative stain EM. A 3-µl aliquot containing ∼0.01 mg ml$^{-1}$ of the sample was applied for 15 s onto a carbon-coated 400 Cu mesh grid that had been glow discharged at 20 mA for 30 s, then negatively stained with 2% uranyl formate for 45 s. Data were collected using a FEI Tecnai Spirit electron microscope operating at 120 kV, with an electron dose of ∼30 e$^-$ Å$^{-2}$ and a magnification of ×52 000 that resulted in a pixel size of 2.05 Å at the specimen plane. Images were acquired with a Tietz 4k × 4k TemCam-F416 CMOS camera using a nominal defocus of 1,000 nm and the Leginon package[66]. The nanoparticles were picked automatically using DoG Picker and put into a particle stack using the Appion software package[67]. Reference-free, 2D class averages were calculated using particles binned via the iterative msa/mra Clustering 2D Alignment[68] and IMAGIC software systems[69], and sorted into classes.

**Enzyme-linked immunosorbent assay.** Each well of a Costar 96-well assay plate (Corning) was first coated with 50 µl PBS containing 0.2 µg of V1V2 antigen. The plates were incubated overnight at 4 °C and then washed five times with wash buffer (PBS containing 0.05% Tween 20). Each well was then coated with 150 µl of a blocking buffer consisting of PBS, 20 mg ml$^{-1}$ blotting-grade blocker (Bio-Rad) and 5% fetal bovine serum. The plates were incubated with the blocking buffer for 1 h at room temperature and then washed five times with wash buffer. Next, the apex-directed bNAbs were diluted in the blocking buffer to a maximum concentration of 2 µg ml$^{-1}$, followed by a fivefold dilution series. For each antibody dilution, a total of 50 µl was added to the appropriate wells. Each plate was incubated with the primary antibodies for 1 h at room temperature and then washed five times with wash buffer. Next, a 1:5,000 dilution of goat anti-human IgG antibody (Jackson ImmunoResearch Laboratories, Inc.) was made in the wash buffer, with 50 µl of this diluted secondary antibody added to each well. The plates were incubated with the secondary antibody for 1 h at room temperature and then washed five times with wash buffer. Finally, the wells were developed with 50 µl of TMB (Life Sciences) for 3–5 min before stopping the reaction with 50 µl of 2 N sulfuric acid, all at room temperature. The resulting plate readouts were measured at a wavelength of 450 nm.

**Biolayer interferometry.** The kinetics of HIV-1 antigen (monomer, trimer and nanoparticle) binding to bNAbs and non-NAbs was measured using an Octet RED96 instrument (fortéBio, Pall Life Sciences). All assays were performed with agitation set to 1,000 r.p.m. in fortéBIO 1 × kinetic buffer. The final volume for all the solutions was 200 µl per well. Assays were performed at 30 °C in solid black 96-well plates (Geiger Bio-One). Antibody (5 µg ml$^{-1}$) in 1 × kinetic buffer was loaded onto the surface of anti-human Fc Capture Biosensors (AHC) for 300 s. A 60-s biosensor baseline step was applied before the analysis of the association of the antibody on the biosensor to the testing antigen in solution for 200 s. A twofold concentration gradient of testing antigen was used in a titration series of six. The dissociation of the interaction was followed for 300 s. Correction of baseline drift was performed by subtracting the averaged shift recorded for a sensor loaded with antibody but not incubated with antigen, or a sensor without antibody but incubated with antigen. The concentrations of soluble monomers, trimers and nanoparticles in this assay were calculated according to the estimated m.w. For example, the m.w. of a gp140 trimer was calculated as m.w. of (gp140 + glycans) × 3, estimated to be ∼530 kDa, whereas the m.w. of a gp140-FR nanoparticle was calculated as m.w. of (gp140-FR + glycans) × 24, estimated to be ∼4696 kDa. Of note, these m.w. values are only approximate and may vary due to the heterogeneity of glycans. Octet data were processed by fortéBio's data acquisition software v.8.1. Experimental data were fitted for V1V2 apex-directed bNAbs using a global fit 1:1 model to determine the $K_D$ values and other kinetic parameters.

**Surface plasmon resonance.** SPR experiments were performed to determine the kinetics for bNAb binding to nanoparticles gp140.664-10aa-FR and gp140.664-E2p. All experiments were performed on a Biacore 3000 optical biosensor equipped with research-grade CM3 sensor chip (Biacore AB, Uppsala, Sweden). The instrument temperature was set at 25 °C for all kinetic analyses. Filtered and degassed 1 × HBS-EP + buffer (Biacore AB) was used as running buffer and sample buffer for all analyses. PGT128 IgG was first immobilized onto CM3 sensor chip in flow cells 2 and 4 at an immobilization level about 3,500 RU by NHS/EDC chemistry using Amine Coupling Kit (GE Healthcare), and a control antibody was also immobilized at a similar immobilization level in flow cells 1 and 3 as reference. To prepare the surface for kinetic analysis, gp140.664-E2p or gp140.664-10aa-FR1 particle was diluted to the concentration of 1 µg ml$^{-1}$ in running buffer and the particle solution was injected over all four flow cells at a flow rate of 5 µl min$^{-1}$ for 60 min. The chip surface was then washed extensively by running buffer at a flow rate of 100 µl min$^{-1}$ until the baseline was stabilized. A custom wizard application was devised to conduct the kinetic analysis, including the following elements: (1) detection mode was varied by cycle, in which flow cell 2-1 (Fc2-1) was used for Fab analysis and flow cell 4-3 (Fc4-3) was used for IgG detection; (2) flow rate of 30 µl min$^{-1}$ was set for all experiments; (3) a 10-min wash step was used to stabilize the baseline before sample injection; (4) a 5-min sample kinetic injection with a 20-min dissociation phase was set and sample injection was varied by cycle, in which Fc 2-1 was for Fab injection and Fc 4-3 was for IgG injection; (5) an additional wash step of 20-min was inserted to wash off any bound antibody. To ensure all bound antibody has been washed away from the chip surface and the baseline was stabilized, an extensive wash step running at 100 µl min$^{-1}$ was set separately between each concentration analysis. Samples (PGT151 Fab, PGT151 IgG, PGDM1400 Fab and PGDM1400 IgG) were prepared freshly from the stock in running buffer at the concentration of 25, 50, 100, 200 and 400 nM, and were injected individually over the designated chip surface according to the custom wizard application. A few samples were injected in duplicates to ensure the binding is reproducible. Several random buffer 'blank' injections were also conducted over different surfaces (Fc 2-1 or Fc 4-3) as reference before, between and after antibody sample injections. Both data processing and kinetic fitting were performed using BIAevaluation ver 4.1 (Biacore AB). The binding responses from each sample were double-referenced, that is, subtract the response over the reference surface and subtract the average response of the 'blank' injection from the sample binding responses. The association and dissociation phase data sets were merged together to globally fit the responses. The data were fitted using a 1:1 Langmuir binding model for both Fab and IgG samples. The association and dissociation rate constants ($k_a$ and $k_d$) and the equilibrium dissociation constant ($K_D$) were calculated (Table 1).

**B-cell activation assay.** WEHI231 cells expressing a doxycyclin-inducible form of VRC01 BCR[62] were maintained in advanced DMEM (Gibco), supplemented with

10% FCS, Pen/Strep antibiotics and $2 \mu g \, ml^{-1}$ Puromycin (Gibco). Cells were treated overnight in $1 \mu g \, ml^{-1}$ doxycyclin (Clontech) to induce human BCR expression. After loading with Indo-1 (Molecular Probes) at $1 \mu M$ for 1 h at 37 °C, washed cells were stimulated with the indicated agents at a concentration of $10 \mu g \, ml^{-1}$: anti-human Ig κ-chain F(ab')2 (Southern Biotech), anti-mouse IgM (Jackson ImmunoResearch), HR1 redesign 1 (gp140 trimer), gp120Sht-FR (gp120-FR nanoparticle), gp120Sht-E2p (gp120-E2p nanoparticle) or gp140.664-10aa-FR (gp140-FR nanoparticle). Calcium mobilization was assessed on a LSR II flow cytometer (BD). After each run cells were stimulated with 1 μl of $1 \mu g \, ml^{-1}$ ionomycin (Sigma) to verify indo loading.

**Data availability.** The authors declare that all data supporting the findings of this study are available within the article and its Supplementary Information files.

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

## Acknowledgements

EM data were collected at the Scripps Research Institute EM Facility. This work was supported by the International AIDS Vaccine Initiative Neutralizing Antibody Center and CAVD, by the Center for HIV/AIDS Vaccine Immunology and Immunogen Discovery (CHAVI-ID UM1 AI00663) (A.B.W., I.A.W. and J.Z.), by the HIV Vaccine Research and Design (HIVRAD) programme (P01 AI110657) (A.B.W. and I.A.W.), by the Joint Center of Structural Genomics (JCSG) funded by the NIH NIGMS, Protein Structure Initiative (U54 GM094586) (I.A.W.), AI073148 (D.N.) and AI084817 (I.A.W. and A.B.W.).

## Author contributions

L.H., N.d.V., I.A.W., D.N., A.B.W. and J.Z. designed research. L.H., N.d.V., C.D.M., N.V., T.C.T., P.A., L.K., B.Z. and J.Z. performed experiments. D.S. and D.R.B. provided bNAbs PG9, PG16, PGT145, PGDM1400, PGT121, PGT128, PGT135 and PGT151, as well as non-NAbs b6, F105 and 447-52D. L.H., N.d.V., C.D.M., N.V., T.C.T., P.A., B.Z., I.A.W., A.W.B. and J.Z. analysed the data. L.H., N.d.V., C.D.M., N.V., I.A.W., D.N., A.B.W. and J.Z. wrote the paper. All authors were asked to comment on the manuscript. This is TSRI manuscript number 29255.

## Additional information

**Competing financial interests:** The authors declare no competing financial interests.

