## [Peer review file · Nature Communications]

REVIEWERS' COMMENTS:

Reviewer #1 (Remarks to the Author):

As noted in the author's rebuttal letter, two new sets of experiments are included in the revised manuscript to address my major criticisms. The new production/purification scheme for the gp140.644-10aa-FR particles provides reasonable reassurance that at least some the nanoparticle constructs will be amenable to a production scale that will permit animal testing and early phase human trials.

The studies demonstrating efficient BCR activation in a B cell line expressing VRC101 BCRs by nanoparticles displaying three of the gp120/gp140 constructs are also encouraging. However, they only begin to address the issue of immunogenicity. Naive B cells with high avidity broadly neutralizing BCRs resembling VRC101 will rarely be present in humans to prime this type of antibody response. In addition, while greater BCR activation is almost certainly an important contributor to the exceptional ability of virus-like particulate display to induce antigen-specific antibody responses, other activities are also likely to be involved. Potentially important factors not measured in the author's assay include 1) efficient trafficking to lymph nodes, 2) binding to natural low avidity IgM and complement for efficient delivery to follicular dendritic cells, 3) efficient processing by APCs for induction of T helper responses, and potentially 4) preferential activation of IgD BCRs on naïve B cells. Whether the synthetic nanoparticle reported herein have these activities characteristic of virions and non-infection virus-like particles remains to be determined. Based on current theory, it seems more likely than not that they will.

It is clear that the authors intend to publish the manuscript here, or elsewhere, without immunogenicity data. So we are left to make a decision on its merits with little experimental data with which to judge between the possibility that the study is a major step in the development of an effective HIV vaccine or just another set of HIV vaccine candidates that bind broadly neutralizing antibodies, but fail to efficiently generate them. Perhaps the best way to sum up the study is that it has provided the field with some interesting and novel HIV vaccine candidates that very much merit detailed immunogenicity analyses.

All of my minor points were adequately addressed.

Reviewer #2 (Remarks to the Author):

The revised manuscript addresses my comments/concerns. A minor point: the authors should clarify whether or not during the ELISA and BLI assays, the env concentrations of soluble monomers, trimers, particle-associated are standardized. If however, such standardization was not performed then the conclusions regarding any differential exposure of particular epitopes between the various Env forms (monomers vs. particles, for example) may not be accurate.

Point-by-Point Response to Review Comments

Reviewer 1:

General comment 1: I largely agree with the author's statement that "self-assembling nanoparticles with sufficient size and proper surface spacing that mimic should be a high priority in the future development of HIV-1 vaccines". The manuscript reports an extensive effort to generate this type of vaccine candidates, involving three distinct nanoparticle platforms and display of three different env polypeptides. Significantly, the manuscript reports the first time that native-like trimeric forms of V1V2 and gp120 subdomains have been displayed on a nanoparticle. However, there are two major limitations to the study.

Response:

We thank the reviewer for his/her careful reading of the manuscript and appreciate the reviewer's helpful comments. As stated in the manuscript and agreed upon by the reviewer, the design and *in vivo* testing of nanoparticle immunogens should be a high priority in the future HIV-1 vaccine development. However, little is known about what nanoparticle platforms can be used to display trimeric HIV-1 antigens and what properties can be expected from such chimeric nanoparticles (e.g. expression, structure, antigenicity, and immunogenicity), which have hindered an objective assessment of their potential as vaccine candidates. As noted by both reviewers, our manuscript presents an extensive effort to develop nanoparticle vaccine candidates involving three particle platforms and three Env antigens. It was our hope that this work would spur on future studies to develop new nanoparticles and to investigate the immunogenicity of nanoparticle constructs in this and forthcoming studies. We also thank the reviewer for noting the significance of the V1V2 and gp120-presenting nanoparticles, which may provide promising immunogens to target bNAbs epitopes encoded by these two Env domains. To address the reviewer's comments, we have included new experimental data on nanoparticle production and B cell activation assay, and have substantially revised the manuscript.

General comment 2: From a technical perspective, these nanoparticle platforms are not novel and have been previously used for trimer display. In addition, assembly of the recombinant subunits generated here seems rather suboptimal. The LS fusions were essentially assembly deficient and, based on the SEC results, only a minor fraction of the ferritin and E2p recombinant subunits assembled correctly. Granted this is a proof of concept study that was not intended to generate an optimal production strategy. Nevertheless, yields and purity of the various nanoparticles are not presented and so there remains some concern as to whether any of the candidates can be produced on an industrial scale.

Response:

We agree with the reviewer that the 24-meric ferritin has been used to display the SOSIP trimer (Sliepen et al., *Retrovirology* 2015). However, the use of VLP-size nanoparticles (e.g. 60-meric E2p) to display trimeric HIV-1 antigens is novel and has not been demonstrated. Although the LS designs were assembly-deficient, we felt that it was imperative to report these results because they provided critical insights into various aspects of nanoparticle design and characterization. First, our results highlighted the importance of surface spacing for the particle display of large, complex HIV-1 antigens. The 60-meric LS particle has a diameter of 14.8 nm, which is only slightly larger than the 24-meric ferritin, which has a diameter of 12.2 nm (Fig. S3a); Second, LS has been successfully used to display an engineered outer domain (eOD) (Jardine et al., *Science* 2013). However, our results illustrated the problems associated with this otherwise desirable nanoparticle platform for displaying trimeric HIV-1 antigens, suggesting that further engineering and optimization may be required for the LS nanoparticle; Third, with the failed LS designs and the successful ferritin and E2p designs, we were able to define a proper range of surface spacing that would allow trimeric HIV-1 antigens to be displayed on a nanoparticle surface. We have now included these points in Discussion, on page 19, in the revised manuscript.

There was a small misinterpretation regarding the purity of gp140-E2p nanoparticles. As shown by the SEC profile (Fig. 6b), the GNL-purified material showed a notable high peak corresponding to the well-formed gp140-E2p nanoparticles. The SEC trace of 2G12-purified material was used merely to confirm the results from GNL purification. The reduced nanoparticle peak after 2G12 purification was due to the limited ability of the 2G12 column to capture large E2p nanoparticles. Given the intrinsic purity of gp140-E2p nanoparticles, as indicated by negative-stain EM, 2G12 purification is not necessary. To avoid any confusion, we have now removed the SEC trace of 2G12-purified gp140-E2p from Fig. 6b.

We have identified several approaches to improve the yield of nanoparticle expression. For example, the ExpiCHO™ transient expression system from Thermo Fisher was reported to increase the IgG expression by 25- to 160-fold compared to HEK293 F. We have tested the expression of a representative gp140 nanoparticle (gp140.664-10aa-FR) in ExpiCHO™ cells. The secreted proteins were harvested two weeks after transient transfection and purified using a 2G12 affinity column and SEC on a Superose 6 10/300 GL column. The purified nanoparticle sample was further analyzed by negative-stain EM and BLI using a panel of 5 bNAbs and non-NAbs. The results were rather remarkable. First, 200mL ExpiCHO expression produced a nanoparticle peak in the SEC profile with an absolute UV₂₈₀ absorbance value that is two-fold higher than that obtained from 2L HEK293 F expression. Since the 2G12 flow-through still contained a nanoparticle fraction, we estimated a total ≥20-fold increase in nanoparticle yield; Second, the ExpiCHO-produced sample showed a single nanoparticle peak in the SEC profile, whereas the HEK293 F-produced sample contained a large fraction of trimers, suggesting a substantially improved nanoparticle purity; Third, negative-stain EM revealed a higher efficiency of particle assembly for the ExpiCHO-produced sample, as indicated by the 2D class images; Fourth, the ExpiCHO-produced sample showed significantly improved binding to bNAbs, such as apex-directed PGDM1400, CD4bs-directed VRC01, and gp120-gp41-directed PGT151, relative to the HEK293 F-produced nanoparticle and gp140 trimer. For non-NAb F240, which targets the immunodominant gp41 epitope, the ExpiCHO-produced nanoparticle showed no binding compared to a residual and visible binding observed for the HEK293 F-produced nanoparticle and gp140 trimer, respectively. Taken together, our new experimental data provided convincing evidence that the gp140 nanoparticles can be produced with high yield and high quality, thus paving the way for industrial scale production and large-scale animal studies. We have now included these results in a new Figure (Fig. 7a) and added a new section in the revised manuscript, on page 16.

General comment 3: Second, and much more importantly, there is no immunogenicity data. It is often stated that antigenicity does not equal immunogenicity, and this proposition has been amply demonstrated in the case of HIV vaccines. Despite the extensive physical and antigenicity analyses, I could not decipher which of constructs are the most promising candidates and the authors do not hazard a guess. So, although theoretic arguments can be made as to why one or more of these candidates may be an advance in HIV vaccine development, there is no way to reasonably predict which, if any, will be a superior vaccine in inducing a higher titer and/or more broadly neutralizing antibody response than existing candidates such as the previously reported SOSIP trimers. It must be acknowledged that a proper investigation of the immunogenicity of all these constructs would probably generate enough data for several additional papers. Perhaps the immunogenicity analysis in this paper could be limited to what the investigators consider their lead candidate and thereby give the reader some indication if advances in HIV vaccine development have actually been made. In summary, there is no doubt that the manuscript merits publication. The question is whether a technically state of the art study in nanoparticle construction that does not address the central issue of the overall project deserves to be published in a high impact journal.

Response:

We agree with the reviewer that immunogenicity will be the most important criterion in the assessment of nanoparticles presented in this manuscript. However, as also noted by the reviewer, a proper investigation of the immunogenicity of all nanoparticle constructs would probably generate enough data for several

additional papers. In addition, critical factors such as animal model, adjuvant, and regimen will have to be carefully considered in the experimental design to achieve optimal results. A proper investigation of immunogenicity will likely require significant efforts and resources that are beyond the scope of this study. We are currently producing nanoparticles and setting up collaborations for a systematic assessment of immunogenicity for the nanoparticles with desirable biophysical, structural, and antigenic properties. The immunogenicity data will be reported in our follow-up papers.

Notwithstanding, we have sought to assess the immunogenic potential of these nanoparticles using other well-established methods. We have conducted Ca^{2+} flux experiments to assess the ability of nanoparticles gp120-FR, gp120-MP1, and gp140-FR (all with sufficient yields) to stimulate B cells expressing broadly neutralizing antibodies (bNAbs) as cognate receptors. This assay has been successfully used to assess a germline-targeting CD4-binding site (CD4bs) nanoparticle immunogen (Jardine *et al.*, Science 2013). Significantly, when tested using B cells carrying VRC01 BCRs, all three nanoparticles showed notably higher triggering signals than the gp140 trimer, suggesting that they are promising multivalent vaccine candidates and may engage B cells bearing bNAb characteristics *in vivo*. The B cell data also filled the gap between the biophysical/structural/antigenic analyses and *in-vivo* studies. Furthermore, the effective B cell stimulation by two gp120 nanoparticles suggested that perhaps all nanoparticles validated in this study are worth testing in animal immunization rather than only one based on the perceived ranking (e.g. gp140 > gp120 > V1V2). And some nanoparticles may be more advantageous than others in a certain context, e.g. in the context of epitope-focused vaccine, a V1V2 nanoparticle may elicit apex-directed B cell responses, whereas a gp140 nanoparticle most likely will induce diverse B cell responses to a wide spectrum of neutralizing and non-neutralizing epitopes on the Env.

We have now included the B cell activation data in Fig.7 (Fig. 7c) and added a new section in the revised manuscript, on page 17.

(1). P.3, line 1: "must" seem a bit strong here.

Response:

We have now changed this to "A critical goal of vaccine development for human immunodeficiency virus type-1 (HIV-1) is to induce broadly neutralizing antibodies (bNAbs) in naïve individuals".

(2). P. 4, top: it would be helpful to also indicate the diameter of the three particles in the introductory text.

Response:

We have now indicated the diameters of three nanoparticles in the Introduction section, on page 4.

(3) P. 11, bottom: "showed notably reduced binding" compared to what?

Response:

We have now changed it to "Compared to the individual trimer, this gp140 nanoparticle showed notably reduced binding by ELISA to apex-directed bNAbs PG9 and PGT145, and to a bNAb directed to the gp120-gp41 interface, PGT151"

(4) P. 12, line 7: Do you imagine that "well separated trimer spikes" are an advantage or a disadvantage? One could theorize that it could be an advantage in providing improved access to membrane proximal p41 epitopes. However, it might be a disadvantage in moving the apical V1/V2 determinants too far apart to promote "virus-like" cross-linking of cognate BCRs. It could also increase flexibility that could be a disadvantage in BCR activation. In the Intro or the Discussion, it might be

worthwhile to address the potential advantages and disadvantages of the various nanoparticle constructs in generating bnAbs. As it stands, the rationale for generating each of them is not entirely clear.

Response:

We thank the reviewer for this insightful comment.

We have performed two new experiments that may provide some insights into this issue. We first utilized SPR to examine whether IgG can cross-interact with two adjacent gp140 trimers on the nanoparticle surface. For gp140.664-E2p, which has a larger surface spacing, faster on-rates and slower off-rates were observed for both PGDM1400 and PGT151 IgGs relative to their cognate Fabs, suggesting that the two arms of an IgG can interact with two adjacent trimers on the nanoparticle surface. By contrast, gp140.664-10aa-FR, which has a smaller size but greater curvature, did not show such patterns. We then conducted B cell activation assay to test three nanoparticles for their ability to trigger B cells with bNAb VRC01 expressed as cognate receptors. In this experiment, all three nanoparticles displayed stronger B cell triggering signals than the individual gp140 trimer, suggesting “virus-like” cross-linking of cognate BCRs due to the avidity effect.

We have now reported these results in a new section (pages 16-17) and a new Figure (Figs. 7b and 7c). We have also expanded the discussion, on page 19, to address advantages and disadvantages of different nanoparticle platforms, and included the implications of assembly-deficient LS designs for nanoparticle design and characterization.

(5) P. 12 bottom: it would be helpful to show the SEC profile for SOSIP-ferritin for comparison.

Response:

We have now included two SEC profiles of SOSIP-ferritin for comparison (Fig. S4a). Of note, the SEC profiles of SOSIP-ferritin protein showed a monomer peak, which is absent in the SEC profiles of ferritin nanoparticles presenting a stabilized gp1410 trimer (HR1 redesign 1, described in our companion paper), suggesting that SOSIP-ferritin contains more misfolded gp140-ferritin monomers. This pattern has been consistently observed in multiple SOSIP-ferritin and gp140.664-FR productions.

We have now reported this result on page 12 and also in Fig. S4a.

(6) P. 13, 2nd to last line: Shouldn't this be designated "affinity" rather than "avidity" since it describes the binding of a MoAb antibody? Or do you think that there is bivalent binding? In that regard, it would be interesting to calculate which constructs would be able to allow bivalent binding of known bnAbs. Bivalent binding would presumably allow initial activation of lower affinity BCRs on naïve B cells and so could potentially increase Ab diversity.

Response:

We thank the reviewer for this insightful comment. Please see our detailed response to comment (4).

(7) P. 16, middle: "gp140.664-E2p showed reduced recognition by the bnAbs" relative to what, free trimers or the ferritin conjugates?

Response:

To clarify this statement, we have now changed it to “gp140.664-E2p showed reduced recognition by bnAbs relative to individual trimer and gp140-ferritin nanoparticles”.

(8) Discussion, 2nd sentence: please consider changing "a modern age" to "a new age". "modern" has a bit too much of a pejorative connotation for pass efforts for my taste.

Response:

We have now changed “modern age” to “new era” in the revised manuscript.

(9) Discussion, line 5: please change "the current focus" to "a current focus". There are reasonable investigators working on HIV vaccine concepts that aren't exclusively based on gp140 trimers.

Response:

We have now changed it to “As gp140 trimer design becomes a focus of current HIV-1 research” in the revised manuscript.

(10) P. 18, top: It seems a bit unfair to note the suboptimal immunogenicity of the SOSIP-ferritin nanoparticles generated in Slieden et al. when no immunogenicity results are reported for the recombinant nanoparticles generated herein.

Response:

Based on the reviewer’s suggestion, we have now removed “immunogenicity” from the statement.

Reviewer 2:

General comments: He and colleagues, present an in-depth study on the design, expression, purification and antigenic characterization of three distinct particle platforms: Ferritin, lumazine synthase (LS) and dihydrolipoyl acyltransferase (E2p) expressing the VIV2 domain of gp120 ('short' and 'longer' versions), gp120 (again two forms), or trimeric stabilized SOSIP gp140s. The authors discuss that based on available studies, multimeric envelope forms are expected to stimulate more robust B cell responses than soluble envelope trimers (such as SOSIP for instance). The stated purpose of the study is to identify a particulate formulation that is easily produced and which expresses multiple copies of envelope, for immunization studies. I believe that this is a valid argument that justifies the performance of the present study. One important consideration is that the envelope molecules present on the surface of such particles, should maintain a native antigenic conformation. The antigenic characterization of envelope discussed here, is based on comparing the relative binding of neutralizing antibodies (whose epitopes are normally expressed on the native trimers) and non-neutralizing HIV antibodies (whose epitopes should be less exposed on the native trimers). I find the study to be timely, thorough and significant. I do not have major concerns or significant criticism.

Response

We appreciate the reviewer’s positive comments and valuable suggestions. Based on both reviewers’ comments, we have included substantial new data on high-yield nanoparticle production, SPR assessment of avidity effect (IgG versus Fab) for two bNAbs, and B cell activation assay in the revised manuscript (Fig. 7 and Results, pages 16-17). We hope that these changes we made have sufficiently addressed both reviewers’ comments and questions.

(1) It is somewhat disconcerting that the epitopes of CD4-binding site antibodies with very limited breadth of neutralization (such as F105 and b6) are exposed on trimeric envelope molecules on the surface of the different particles (in fact the BLI traces suggest that the epitopes of such antibodies and the epitopes of VRC01 may be equally accessible). Previous studies on soluble BG505 trimeric gp140 proteins indicated that epitopes of F105 and b6 were occluded. Isn't this something to be avoided? - In general, it seems that the observations made on the relative expression of the three particle-envelope platforms will depend on the envelope background used (here only 1-2 different envelope backgrounds were used). Is this an accurate assumption?

Response:

We thank the reviewer for this insightful comment on the CD4bs recognition by non-bNAbs.

Both the displayed antigen and the nanoparticle platform may have contributed to this effect. The trimeric gp120 exhibited stronger binding to F105 and b6 (Figs. 2 and 3) than the gp140 trimer displayed on the same nanoparticle (Figs. 5 and 6), likely due to the lack of structural constraints imposed by gp41 and thus greater CD4bs accessibility. The large surface spacing of E2p nanoparticle further enhanced the CD4bs accessibility, and as a result, gp120-E2p exhibited stronger binding to F105 and b6 than gp120-ferritin. Similar patterns were observed for the nanoparticle binding to VRC01, suggesting a similar effect of structural constraints (antigen) and surface spacing (particle) on antibodies targeting the same epitope. It is also worth pointing out that the gp140 trimer used for nanoparticle display – HR1 redesign 1 – has demonstrated improved trimer yield and purity, increased gp41 stability, and decreased F105 binding compared to the SOSIP trimer (Fig. 3 in our companion paper). When this gp140 trimer was displayed on ferritin, the resultant gp140 nanoparticle displayed stronger binding to VRC01 than to F105 (Fig. 5a, column 5 vs. Fig. 5b, column 1). A less visible difference was observed for gp140-E2p (Fig. 6e, column 3 vs. Fig. 6f, column 1). The experimental methods (SPR versus BLI) may have also contributed to the observed binding to F105 and b6. In the previous studies, the SOSIP trimers were immobilized on a Biacore chip and tested for binding in a flow-through solution that contained antibodies, whereas in the current studies, IgGs were immobilized on an Octet sensor and dipped into a shaking solution containing either gp140 trimers or nanoparticles. The more exposed trimer surface in the BLI setup might explain the seemingly ‘enhanced’ binding to non-bNAbs F105 and b6. Using the same BLI protocol, the NFL trimer showed relatively high F105 binding that was reduced to a level similar to ours after F105-based negative selection (Sharma et al., *Cell Rep* 2015). Overall, the HR1-redesigned gp140-ferritin nanoparticle showed the most significant difference in binding to bNAb VRC01 and to non-bNAbs F105 and b6.

In this study, we tested ZM109 and CAP45 for V1V2 nanoparticles and BG505 for gp120 and gp140 nanoparticles. We speculate that the success of particle display may be determined by both the envelope backbone and the displayed antigen type. If the gp140 trimer derived from a particular envelope displays similar purity, stability, structure and antigenicity to that derived from clade-A BG505, it is most likely that all three antigens – V1V2, gp120, and gp140 – derived from this envelope will be compatible with nanoparticle display. In the companion manuscript, we examined the HR1 redesign for diverse envelope backbones and observed significant improvement in terms of trimer yield, purity and stability for clade-A, -B, and -C envelopes. It is reasonable to assume that the trimeric antigens from these envelopes can be successfully displayed on ferritin and E2p nanoparticles.

(2) Fig 1f vs 1g: During ELISA the particles are adsorbed on the plate directly while during BLI the Abs are captured on the chip and dipped into a solution containing the particles. PG9 binds relatively well to ZM109 V1V2 by ELISA but barely by BLI. Is it because of the different orientation of the two assays?

Response:

We agree with the reviewer that different V1V2 orientation in two assays may contribute to the seemingly different binding results. Another possible explanation is that these two assays measure different aspects of antibody-antigen interactions. In the ELISA assay, PG9 was added to the V1V2-coated plate for ~1 hour to allow the binding reaction to reach equilibrium, whereas in the BLI experiment, the kinetics of association between PG9 and V1V2 was measured for 200 s. Although the signal on Octet is relatively low (a maximum of 0.2 nm shift compared to 0.5 for V1V2 nanoparticles), the binding affinity (K_D) was determined to be 102 nM, indicating a reasonable binding between PG9 and monomeric V1V2.

(3) Fig 1c ad pg 7 (2nd paragraph). What do the authors mean when they say that 'V1V2Sht-FR exhibited similar binding kinetics with respect to V1V2 Ext-FR'? the binding affinities for both PG9 and PGDM1400 were lower for the Ext-FR than for the Sht-FR.

Response:

We thank the reviewer for pointing out this confusion. We have substantially revised the V1V2 section to improve the clarity. Now it reads “Of note, V1V2Sht-FR exhibited lower affinities for both bNAbs in comparison to V1V2Ext-FR, suggesting an adversary effect of the shortened V1V2 on the apex structure and antigenicity”.

(4) End of pg7-beginning of pg8: the authors mention that the importance of trimer constraints for HIV-1 neutralization is demonstrated by immunogenicity studies performed in human Ig Knock-in mice in which BG505 SOSIP trimmers but not eOD-nanoparticles elicited Nab responses. That may be true for PGV04 (as the authors correctly point out) but the immunization studies were performed in mice expressing the mutated VH of 3BNC60. Does 3BNC60, like PGV04, require envelope multimerization for binding?

Response:

All VRC01-class bNAbs identified thus far showed preferential use of IgHV1-2*2 germline gene for heavy chain and almost an identical angle of approach when they bound the CD4bs (Wu et al., *Science* 2010; Zhou et al., *Immunity* 2013). Both 3BNC60 and PGV04 belong to the VRC01 class (Scheid et al., *Science* 2011; Wu et al., *Science* 2011). Furthermore, as previously demonstrated, the mature VRC01-class heavy chains can pair with different VRC01-class light chains with intact VRC01-like neutralizing function (Wu et al., *Science* 2011; Zhou et al., *Immunity* 2013; Zhu et al., *PNAS* 2013), indicating that such ‘hybrid’ antibodies must adopt the same angle of approach as the wild-type bNAbs. As noted by the reviewer, the knock-in mice that showed broad NAb responses were expressing mature 3BNC60 heavy chain paired with mouse light chains (Dosenovic et al., *Cell* 2015), which likely retained the same angle of approach towards the native-like trimer as did PGV04 (Lyumkis et al., *Science* 2013). Taken together, 3BNC60 and PGV04 should bind the native-like trimer in a highly similar manner and the correct trimeric context is required for eliciting VRC01-like NAbs in immunization.

(5) On page 13 the authors mention that in contrast to a recent report (Sliepen et al in Retrovirology) , " the fully formed particles have the same "estimated shift" from the trimer band. The gels are run differently here and in that study (the gel on the Sliepen study did not run as long as the gel here). So I am not convinced this statement is accurate.

Response:

We thank the reviewer for noting this critical point. There might be a slight misunderstanding regarding the ‘reference’ from which the band shift on BN gel was estimated. In this study, the shift of the gp140-ferritin band relative to the gp120-ferritin band – not the trimer band – was estimated. Our reasoning was that the gp120-ferritin nanoparticle provides a good reference in the BN-PAGE analysis of gp140-ferritin nanoparticles since the gp140 mass is mainly contributed by gp120 protein and attached glycans. Thus, we expected to see a gp140-ferritin band above the gp120-ferritin band but below the well on the BN gel.

We also thank the reviewer for noting the potential difference in run time. In our study, we have tested several shorter run times for gp140-ferritin nanoparticles. However, even with a 30-min run time (versus recommended 2~3 hours), we still observed bands on the BN gel with minimal material remaining in the wells, which were shown as dark lines on the top of the BN gel. By contrast, all SOSIP-ferritin particles remained in the BN well in the previous study (Sliepen et al., *Retrovirology* 2015). Perhaps experimental factors other than the running time were involved.

To address the reviewer’s comment, we have now modified it to “All three HR1-redesigned gp140-FR constructs showed high m.w. bands in BN-PAGE corresponding to fully assembled nanoparticles (**Fig. 4e**), which are consistent with the SEC profiles (**Fig. 4d**) and the expected shift relative to the gp120-FR BN bands (**Fig. 2d**), but in contrast to the previous work showing SOSIP-FR particles at the top of the BN gel”, on page 12, in the revised manuscript.

(6) It is difficult to tell from the SEC traces in all the figures, what proportion of envelope was expressed and purified as particles and what proportion was not.

Response:

We thank the reviewer for this helpful comment. We have modified all figures and supplementary figures containing SEC traces to indicate the nanoparticle portion of the expressed/purified protein with dashed box and provided the volume range (e.g. X-Y mL) in figure legends.

(7) Pg10. EM analysis of E2p expressing envelope also showed 2D averages lacking the gp120 spikes. It is hypothesized (pg 11) that it is possible that three gp120s around each threefold axis fail to assemble into stable trimers. I don't see how antibody-binding analysis can clarify this. Particles may have both trimeric and non-trimeric envelope forms on them.

Response:

We agree with the reviewer that these nanoparticles may present both trimeric and non-trimeric gp120 on the surface. The binding to apex-directed bNAbs will prove that at least some gp120s have formed native-like trimeric apex as observed for BG505 SOSIP.664 gp140 trimer, but cannot rule out the possibility that a fraction of displayed gp120s have remained non-trimeric. Based on the reviewer's comment, we have now modified the manuscript, on page 10, to clarify this point. Now it reads "It was unclear from EM whether some, if not all, gp120s had formed trimeric spikes".

(8) Details on how ELISA was performed are absent from the methods section.

Response:

We have now included the ELISA method in the revised manuscript, on page 22.

Reviewers' Comments:

Reviewer #1 (Remarks to the Author)

I largely agree with the author's statement that "self-assembling nanoparticles with sufficient size and proper surface spacing that mimic should be a high priority in the future development of HIV-1 vaccines". The manuscript reports an extensive effort to generate this type of vaccine candidates, involving three distinct nanoparticle platforms and display of three different env polypeptides. Significantly, the manuscript reports the first time that native-like trimeric forms of V1V2 and pg120 subdomains have been displayed on a nanoparticle. However, there are two major limitations to the study. From a technical perspective, these nanoparticle platforms are not novel and have been previously used for trimer display. In addition, assembly of the recombinant subunits generated here in seems rather suboptimal. The LS fusions were essentially assembly deficient and, based on the SEC results, only a minor fraction of the ferritin and E2p recombinant subunits assembled correctly. Granted this is a proof of concept study that was not intended to generate an optimal production strategy. Nevertheless, yields and purity of the various nanoparticles are not presented and so there remains some concern as to whether any of the candidates can be produced on an industrial scale. Second, and much more importantly, there is no immunogenicity data. It is often stated that antigenicity does not equal immunogenicity, and this proposition has been amply demonstrated in the case of HIV vaccines. Despite the extensive physical and antigenicity analyses, I could not decipher which of constructs are the most promising candidates and the authors do not hazard a guess. So, although theoretic arguments can be made as to why one or more of these candidates may be an advance in HIV vaccine development, there is no way to reasonably predict which, if any, will be a superior vaccine in inducing a higher titer and/or more broadly neutralizing antibody response than existing candidates such as the previously reported SOSIP trimers. It must be acknowledge that a proper investigation of the immunogenicity of all these constructs would probably generate enough data for several additional papers. Perhaps the immunogenicity analysis in this paper could be limited to what the investigators consider their lead candidate and thereby give the reader some indication if advances in HIV vaccine development have actually been made. In summary, there is no doubt that the manuscript merits publication. The question is whether a technically state of the art study in nanoparticle construction that does not address the central issue of the overall project deserves to be published in a high impact journal.

Specific Points:

1. P.3, line 1: "must" seem a bit strong here.
2. P. 4, top: it would be helpful to also indicate the diameter of the three particles in the introductory text.
3. P. 11, bottom: "showed notably reduced binding" compared to what?
4. P. 12, line 7: Do you imagine that "well separated trimer spikes" are an advantage or a disadvantage? One could theorize that it could be an advantage in providing improved access to membrane proximal p41 epitopes. However, it might be a disadvantage in moving the apical V1/V2 determinants too far apart to promote "virus-like" cross-linking of cognate BCRs. It could also increase flexibility that could be a disadvantage in BCR activation. In the Intro or the Discussion, it might be worthwhile to address the potential advantages and disadvantages of the various nanoparticle constructs in generating bnAbs. As it stands, the rationale for generating each of them is not entirely clear.
5. P. 12 bottom: it would be helpful to show the SEC profile for SOSIP-ferritin for comparison.
6. P. 13, 2nd to last line: Shouldn't this be designated "affinity" rather than "avidity" since it describes the binding of a MoAb antibody? Or do you think that there is bivalent binding? In that regard, it

would be interesting to calculate which constructs would be able to allow bivalent binding of known bnAbs. Bivalent binding would presumably allow initial activation of lower affinity BCRs on naïve B cells and so could potential increase Ab diversity.

7. P. 16, middle: "gp140.664-E2p showed reduced recognition by the bNAbs" relative to what, free trimers or the ferritin conjugates?

8. Discussion, 2nd sentence: please consider changing " a modern age" to "a new age". "modern" has a bit too much of a pejorative connotation for pass efforts for my taste.

9. Discussion, line 5: please change "the current focus" to "a current focus". There are reasonable investigators working on HIV vaccine concepts that aren't exclusively based on gp140 trimers.

10. P. 18, top: It seems a bit unfair to note the suboptimal immunogenicity of the SOSIP-ferritin nanoparticles generated in Sliepen et al. when no immunogenicity results are reported for the recombinant nanoparticles generated herein.

Reviewer #2 (Remarks to the Author)

He and colleagues, present an in-depth study on the design, expression, purification and antigenic characterization of three distinct particle platforms: Ferritin, lumazine synthase (LS) and dihydrolipoyl acyltransferase (E2p) expressing the V1V2 domain of gp120 ('short' and 'longer' versions), gp120 (again two forms), or trimeric stabilized SOSIP gp140s. The authors discuss that based on available studies, multimeric envelope forms are expected to stimulate more robust B cell responses than soluble envelope trimers (such as SOSIP for instance). The stated purpose of the study is to identify a particulate formulation that is easily produced and which expresses multiple copies of envelope, for immunization studies. I believe that this is a valid argument that justifies the performance of the present study. One important consideration is that the envelope molecules present on the surface of such particles, should maintain a native antigenic conformation. The antigenic characterization of envelope discussed here, is based on comparing the relative binding of neutralizing antibodies (whose epitopes are normally expressed on the native trimers) and non-neutralizing HIV antibodies (whose epitopes should be less exposed on the native trimers). I find the study to be timely, thorough and significant. I do not have major concerns or significant criticism. I do however hope that the authors could comment on the following:

- It is somewhat disconcerting that the epitopes of CD4-binding site antibodies with very limited breadth of neutralization (such as F105 and b6) are exposed on trimeric envelope molecules on the surface of the different particles (in fact the BLI traces suggest that the epitopes of such antibodies and the epitopes of VRC01 may be equally accessible). Previous studies on soluble BG505 trimeric gp140 proteins indicated that epitopes of F105 and b6 were occluded. Isn't this something to be avoided?

- In general, it seems that the observations made on the relative expression of the three particle-envelope platforms will depend on the envelope background used (here only 1-2 different envelope backgrounds were used). Is this an accurate assumption?

- Fig 1f vs 1g: During ELISA the particles are adsorbed on the plate directly while during BLI the Abs are captured on the chip and dipped into a solution containing the particles. PG9 binds relatively well to ZM109 V1V2 by ELISA but barely by BLI. Is it because of the different orientation of the two assays?

- Fig 1c ad pg 7 (2nd paragraph). What do the authors mean when they say that 'V1V2Sht-FR exhibited similar binding kinetics with respect to V1V2 Ext-FR'? the binding affinities for both PG9 and PGDM1400 were lower for the Ext-FR than for the Sht-FR.

- End of pg7-eginning of pg8: the authors mention that the importance of trimer constraints for HIV-1 neutralization is demonstrated by immunogenicity studies performed in human Ig Knock-in mice in which BG505 SOSIP trimers but not eOD-nanoparticles elicited Nab responses. That may be true for PGV04 (as the authors correctly point out) but the immunization studies were performed in mice expressing the mutated VH of 3BNC60. Does 3BNC60, like PGV04, require envelope multimerization

for binding?

- On page 13 the authors mention that in contrast to a recent report (Sliepen et al in Retrovirology) , " the fully formed particles have the same "estimated shift" from the trimer band. The gels are ran differently here and in that study (the gel on the Sliepen study did not run as long as the gel here). So I am not convinced this statement is accurate.
- It is difficult to tell from the SEC traces in all the figures, what proportion of envelope was expressed and purified as particles and what proportion was not.
- Pg10. EM analysis of E2p expressing envelope also showed 2D averages lacking the gp120 spikes. It is hypothesized (pg 11) that it is possible that three gp120s around each threefold axis fail to assemble into stable trimers. I don't see how antibody-binding analysis can clarify this. Particles may have both trimeric and non-trimeric envelope forms on them.
- Details on how ELISA was performed are absent from the methods section.